# INFORMATION-THEORETIC MEMBERSHIP INFERENCE FOR GRANULAR QUANTIFICATION OF MEMORIZATION

**Jiashu Tao[†], and Reza Shokri[†‡]**
[†] National University of Singapore, [‡] Google Research
{jiashut, reza}@comp.nus.edu.sg

## ABSTRACT

Machine learning models are known to leak sensitive information, as they inevitably memorize (parts of) their training data. This risk is amplified for large language models (LLMs), which are trained on massive corpora and therefore create a more urgent need for privacy assessment prior to release. The standard method to quantify privacy is via membership inference attacks, where the state-of-the-art approach is the Robust Membership Inference Attack (RMIA). In this paper, we introduce **InfoRMIA**, a principled information-theoretic formulation of membership inference that consistently outperforms RMIA across benchmarks while improving computational efficiency. Moving beyond attack performance alone, we show that treating sequence-level membership inference as the gold standard obscures how memorization manifests in LLMs. To address this limitation, we propose a fine-grained memorization assessment framework based on token-level signals, with InfoRMIA serving as its algorithmic backbone. Our approach identifies which tokens within generated outputs are memorized, localizing privacy leakage from sequences to individual tokens. This framework enables more precise analysis of LLM memorization and potentially targeted mitigation strategies such as exact unlearning.

## 1 INTRODUCTION

In the past decade, researchers have shown that machine learning (ML) models inevitably memorize parts of their training data (Feldman, 2020; Feldman & Zhang, 2020). Memorized data, once identified and extracted, can pose a severe privacy risk. It is increasingly concerning as the contemporary, easily accessible large language models (LLMs) are trained on datasets so large that we are running out of training data (Villalobos et al., 2024). These LLMs have seen nearly all data generated by humans. Even limited memorization by them can translate into significant privacy risks.

The current standard for quantifying privacy is membership inference attacks (MIAs) (Shokri et al., 2017), where the attacker or privacy auditor aims to determine if a given data sample was part of the target model's training set. A stronger attack means the attacker can more accurately distinguish members (training data) from non-members, implying greater information leakage from the model. This ability to separate members from non-members not only signals privacy risk but also raises the possibility of training data reconstruction. It is also closely linked to memorization, as it is the root cause of successful MIAs. Hence, MIAs are widely regarded as the backbone of ML privacy research. The state-of-the-art (SOTA) MIA is the Robust Membership Inference Attack (RMIA) (Zarifzadeh et al., 2024), but its reliance on a separate population dataset, whose size scales linearly with the training set, could be a potential limitation, especially for LLMs.

In the first part of the paper, we thoroughly analyze RMIA, from its formulation to signal computation, and propose a more principled statistical test by casting RMIA's setup as a composite hypothesis testing problem. Our approach can also be interpreted through information theory, where we quantify dominance over population data in bits rather than in sample counts. This transforms the attack signal from discrete to continuous, eliminating sensitivity to the population dataset size. We observe that our new attack, InfoRMIA, consistently outperforms RMIA on tabular, image, and text datasets, while requiring far fewer population samples. Thus, InfoRMIA is a lower-cost, higher-power membership inference attack and establishes a new SOTA.

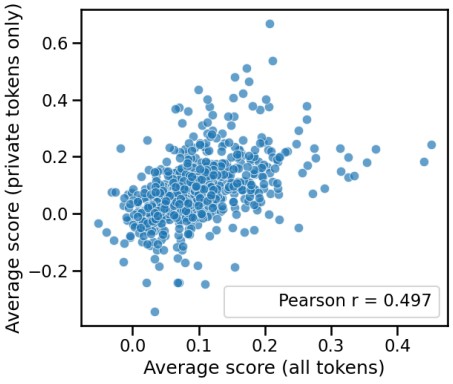

(a) Sequence-level membership scores show weak correlation with the scores of private tokens.

(b) Token-level membership heatmaps obtained from our fine-grained framework reveal that sequences with the highest sequence-level scores primarily memorize non-private tokens.

Figure 1: Sequence-level membership inference can misestimate privacy leakage, as sensitive information is often localized to a small subset of private tokens. The two subfigures illustrate that sequence-level auditing results may diverge substantially from token-level privacy assessments enabled by our framework. Results are obtained using InfoRMIA on GPT-2 models fine-tuned on the ai4privacy dataset (Appendix F.2), which provides privacy masks indicating private token locations.

Although MIAs are the gold standard in quantifying privacy, they follow a strict setup defined by the membership inference game (Yeom et al., 2018; Ye et al., 2022; Zarifzadeh et al., 2024), which falls short in quantifying true information leakage (Tao & Shokri, 2025), especially for LLMs. The current privacy quantification setup for LLMs largely inherit designs developed for MLPs and CNNs: perform MIA on a set of member and non-member sequences. However, transformers operate sequentially, generating predictions token by token. Assigning a single membership label to an entire sequence compresses rich token-based information into a single bit, losing granularity and analogous to a lossy compression (Figure 1a).

To address this limitation, we propose a token-level MIA framework for fine-grained analysis of memorization and information leakage in LLMs. We motivate token-level analysis for three key reasons. First, each token completion corresponds to a single prediction step in autoregressive transformers, making token-level evaluation naturally aligned with model behavior and the definition of membership inference. Second, sequence-level membership scores are inherently aggregated from token-level signals, making them less semantically meaningful for privacy assessment. For example, memorized facts may yield high sequence-based membership scores despite not leaking private information. Third, we argue that private information in a sentence is usually contained in a few words/tokens. Measuring the average memorization of the entire sequence mainly with common words leads to inaccurate privacy assessment (Figure 1b). Token-level analysis can narrow the focus to truly sensitive components, enabling more accurate privacy quantification. By pinpointing the information leakage to individual tokens and words, we can potentially protect privacy more effectively by performing targeted machine unlearning, which would prevent unlearning useful information from non-private texts, while surgically removing the memorization of private information.

In summary, this paper makes the following contributions: (1) we propose InfoRMIA, a principled and efficient improvement over RMIA that achieves new SOTA performance; and (2) we introduce a token-level privacy assessment framework, which offers finer-grained insights into memorization and leakage in LLMs, while achieving strong membership inference capabilities.

## 2 RELATED WORK

### 2.1 MEMBERSHIP INFERENCE

Membership inference (Shokri et al., 2017) is a class of inference attacks that aims to determine if any given point query $x$ is part of the training set of a machine learning model $\theta$. Since its inception, there has been significant progress in the inference strategies. Shokri et al. (2017) trained shadow models to predict the membership label directly. Yeom et al. (2018) proposed a simpler

approach that uses the loss values as the membership signal. To achieve higher inference accuracy, researchers have proposed multiple reference model-based membership inference tests to calibrate the raw signal on the target query. Ye et al. (2022) trained a set of reference models on the population dataset, and counted the number of them with lower probability on the target $x$. Carlini et al. (2022) constructed reference models that train with or without the target point to simulate two distributions of model outputs on the target $x$: the IN and OUT distributions. Assuming Gaussians, the attacker computes the likelihood ratios under the two distributions as the membership signal. The state-of-the-art attack, RMIA (Zarifzadeh et al., 2024), improves further upon reference model-based attacks by counting how many similar data points each test point dominates.

Membership inference techniques on CNNs and MLPs can be adapted to work on LLMs, where the goal is to predict if any given *text sequence* is part of the training dataset. Due to the high computation cost to train reference models, LLM-specific MIAs tend to be reference model-free. Carlini et al. (2021) used entropy, or more easily, zlib (Gailly & Adler, 2004) compression, to calibrate sequence-based membership likelihood. Mattern et al. (2023) compared the perplexity gap between the target and neighboring sequences, while (Shi et al., 2024) looked at the tokens with the smallest probability. Duan et al. (2024) published a benchmark, MIMIR, to evaluate LLM MIAs and found that all of these methods perform poorly on pretrained LLMs. Zhang et al. (2024) and Zhang et al. (2025) improved upon the methods in MIMIR by incorporating additionalx calibration.

## 2.2 MEMORIZATION

Memorization of machine learning models is defined in a leave-one-out fashion by Feldman (2020). Due to its prohibitively high computation cost on LLMs, many alternative definitions have been proposed. So far, verbatim memorization (Carlini et al., 2021; 2023), which means the output sequence **exactly** matches one of the training sequences, is the most popular notion. If the sequence is generated verbatim when conditioned on a given prompt, the memorization term is called discoverable (Carlini et al., 2023; Nasr et al., 2023) or extractable memorization (Nasr et al., 2023), depending on whether the prompt is crafted by an adversary. Hayes et al. (2025b) introduced their probabilistic variations, considering the stochastic nature of LLMs. There are other memorization notions such as $k$-eidetic (Carlini et al., 2021) and counterfactual memorization (Zhang et al., 2023).

## 3 IMPROVING RMIA WITH AN INFORMATION-THEORETIC INSPIRED TEST STATISTIC

In this section, we first briefly explain the problem statement of membership inference under the current SOTA attack, RMIA (Zarifzadeh et al., 2024). We then introduce an improved version of RMIA, which we call information-theoretic RMIA (InfoRMIA). This new attack is consistently stronger than the original RMIA across all datasets and thus establishes a new state-of-the-art.

### 3.1 MEMBERSHIP INFERENCE IN RMIA'S SETTING

In membership inference, the adversary, or the attacker, aims to infer whether any given query $x$ is part of the training data of the target model $\theta$. The adversary only has access to the model output, which can be treated as $p(x|\theta)$. The adversary is also assumed to be able to train reference models $\Theta$ on datasets drawn from the training data distribution of $\theta$, and to have a population dataset $Z$, which is also drawn from the same underlying data distribution. The formal definition is commonly described as an inference game (Yeom et al., 2018; Ye et al., 2022; Carlini et al., 2022; Zarifzadeh et al., 2024; Tao & Shokri, 2025) (See Appendix A).

### 3.2 THE ORIGINAL ROBUST MEMBERSHIP INFERENCE ATTACKS (RMIA)

Carlini et al. (2022) argue that the optimal way to tackle the membership inference problem is to frame it as a hypothesis testing problem and then apply a likelihood ratio test (LRT). In RMIA, Zarifzadeh et al. (2024) formulate the hypothesis testing setup as:

$$H_0 : \text{The target model } \theta \text{ is trained with one of the data } z \in Z,$$
$$H_1 : \text{The target model } \theta \text{ is trained with the given } x. \tag{1}$$

The original test statistic of the LRT in RMIA can be written as

$$\text{Test Statistic} = p_z \left( \frac{p(\theta|x)}{p(\theta|z)} \geq \gamma \right), \tag{2}$$

where $\theta$ is the target model, $x$ is the target query, $z$ is drawn from a population $Z$ of the same data distribution as the training data, and $\gamma \geq 1$ is a hyperparameter that serves as a threshold.

In simple terms, RMIA counts the **proportion** of "similar" data $z$ that the target $x$ dominates. In practice, the test statistic is written as

$$\text{Test Statistic} = p_z \left( \frac{p(x|\theta)}{p(x)} / \frac{p(z|\theta)}{p(z)} \geq \gamma \right) = \frac{1}{|Z|} \sum_{z \in Z} \mathbb{I} \left( \frac{p(x|\theta)}{p(x)} / \frac{p(z|\theta)}{p(z)} \geq \gamma \right), \tag{3}$$

where $\mathbb{I}(\cdot)$ is the identity function and the first equality follows directly from Bayes' Theorem.

Note that the formulation in Eqn 3 makes the membership score a discrete value whose granularity depends on the size of $Z$. Intuitively, the more $z$ data RMIA uses, the finer the "bins" become, and the more distinguishing and precise the signal gets. Empirically, Zarifzadeh et al. (2024) also reported this relationship between the size of $Z$ and the attack performance. The empirical insight was that using $Z$ of about $10\%$ of the training set size is sufficient. However, for LLMs, even $10\%$ represents an astronomical number of samples.

## 3.3 Info-Theoretic RMIA (InfoRMIA)

Instead of counting how much population *data* the target point dominates, we measure how many *bits* the target point saves in explaining the target model relative to the population data in expectation. That is, we want to measure

$$\mathbb{E}_z \left[ -\log p(\theta|z) \right] - (-\log p(\theta|x)) = \log p(\theta|x) - \mathbb{E}_z \log p(\theta|z) \tag{4}$$

By applying the same Bayesian decomposition in Zarifzadeh et al. (2024) and some basic manipulations, we can obtain the following equivalent formulation of our new test statistic in Eqn 4:

$$\begin{aligned}
\text{Test Statistic} &= \sum_z p(z) \log \left( \frac{p(\theta|x)}{p(\theta|z)} \right) = \sum_z p(z) \log \left( \frac{p(x|\theta)p(z)}{p(z|\theta)p(x)} \right) \\
&= \log \left( \frac{p(x|\theta)}{p(x)} \right) + \sum_z p(z) \log \left( \frac{p(z)}{p(z|\theta)} \right) \\
&= \log \left( \frac{p(x|\theta)}{p(x)} \right) + D_{\text{KL}} \left( p(z) \,||\, p(z|\theta) \right)
\end{aligned} \tag{5}$$

Note that the formulation is only valid when $\sum_z p(z) = 1$[1]. Hence, for an empirical or approximated (in RMIA's case) $\tilde{p}(z)$, we need to normalize it to $\hat{p}(z) = \tilde{p}(z)/\sum_z \tilde{p}(z)$. Similarly, for the last step to hold, we require that $\sum_z p(z|\theta) = 1$. For simplicity, we use $p(z)$ and $p(z|\theta)$ in the rest of the paper to denote the normalized distribution of population data $z$. As this new test statistic is inspired by information theory, we refer to it as *InfoRMIA*. Its pseudocode can be found in Appendix C.

**Interpretation of the test statistic** It is interesting that the test statistic has two parts:

1. $\log \left( \frac{p(x|\theta)}{p(x)} \right)$, which measures the amount of information gain in explaining $x$ given a model $\theta$. This can be seen as the memorization of $x$ by model $\theta$.

2. $D_{\text{KL}} \left( p(z) || p(z|\theta) \right)$, which captures the distributional differences between the base probabilistic distribution of $z$ and that conditioned on model $\theta$. This is reminiscent of generalization analysis, as it reflects the changes in the model's predictive performance on $z$'s.

---

[1] Actually, when attacking **one single fixed** model with a **fixed** population dataset, the attack performance is unchanged even if $\sum_z p(z) \neq 1$. This is because the test statistics would be reduced to $\sum_z p(z) \log p(\theta|x) = C \log p(\theta|x)$, where $C = \sum_z p(z) > 0$ is a constant. Hence, the test statistic would preserve the same total order among all $x$'s log likelihood values.

### 3.4 WHY IS INFORMIA SUPERIOR

Both test statistics of the original and InfoRMIA are principled approaches to solve the same hypothesis testing problem (Eqn 1) derived from the same membership inference game (Appendix A). The original RMIA (Equation 3) performs multiple pairwise tests between $H_1$ and each null hypothesis. Each test requires a threshold $\gamma$. The final score is the proportion of null hypotheses rejected in all the pairwise tests. As mentioned before, this score is inherently discrete, with granularity determined by $|Z|$ and increments of $\frac{1}{|Z|}$.

InfoRMIA does not perform multiple pairwise tests. Instead, it opts for a more systematic approach. Similar to what Tao & Shokri (2025) observed, the scenario described by Equation 1 is a composite hypothesis testing problem. One of the principled solutions is to use Bayes Factor (Tao & Shokri, 2025; Jeffreys, 1939), where we compute the expected log likelihood of the composite null hypothesis by $\mathbb{E}_z \left[ \log p(\theta|z) \right]$. Now it becomes clear that InfoRMIA's test statistic (Equation 4) corresponds to the log of the likelihood ratio when using the Bayes Factor.

Apart from **using a more accurate and established test**, InfoRMIA also supersedes the original RMIA by using a **continuous test statistic** (See Equation 4, 5). This results in significantly higher precision in the membership score and also eliminates the need for the hyperparameter $\gamma$. Since the granularity of the score is no longer dictated by the size of $Z$, InfoRMIA is **much less dependent on a large** $Z$, significantly reducing computational overhead when $|Z|$ is fixed and lowering complexity by a constant factor. Experiment results in Section 6 validate these improvements.

**Factors affecting the gap**   The performance gap between InfoRMIA and RMIA mainly depends on the "niceness" of the distribution of population signals $p(z|\theta)/p(z)$ in RMIA. As the population signal distribution gets more even, the loss of precision in the discretization step (computing the percentile) decreases, thereby narrowing its performance gap with InfoRMIA.

## 4 TOKEN-LEVEL INFORMIA FOR ATTACKING LLMS

We have now justified why InfoRMIA surpasses the original RMIA and becomes the new SOTA attack . We now propose our token-level framework where we can pinpoint information leakage and more truthfully estimate privacy risks with token-level InfoRMIA.

### 4.1 FROM SEQUENCES TO TOKENS

So far, membership inference and privacy risks for LLMs have been defined on the sequence level, i.e., whether a given sequence is a member. The majority of the LLM MIAs aim to compute a score on each sequence (Carlini et al., 2021; Mattern et al., 2023) based on its perplexity. However, delving into the mechanisms of LLMs, we can quickly realize that a sequence is not one single output, but an ordered list of outputs. For example, given a training sequence $\mathbf{x} = \{x_1 x_2 \ldots x_k\}$, the LLM $\theta$ optimizes the losses $\ell(x_2, \theta(x_1)), \ell(x_3, \theta(x_1 x_2)), \ldots, \ell(x_k, \theta(x_1 \ldots x_{k-1}))$. Each sequence is more than one training sample; it resembles a dataset containing $k-1$ training (subsequence, label) pairs. To properly adapt existing MIAs to LLMs, we should treat each token generation step, which "labels" each "prefixal" subsequence (subsequences from the start), as one prediction and compute its membership likelihood. In this way, for any sequence of length $k$, the LLM goes through $k-1$ prediction steps, and we should obtain $k-1$ membership scores. In comparison, the existing framework only computes a single membership signal for each sequence, which is a highly compressed signal that loses rich information at each token position.

The token-level framework also provides a more realistic privacy notion for LLMs. Many researchers have pointed out that the current privacy definition via membership inference is too strict and not comprehensive enough, especially for language data, as it only considers *exact* matches as privacy concerns (Tao & Shokri, 2025; Duan et al., 2024). We believe that the privacy risk of a text sequence primarily resides in the tokens carrying the sensitive information. From an information-theoretic point of view, the total private information in bits can be computed by PrivBits $= \sum_{x \in V_{priv}} -\log p(x) < \sum_x -\log p(x)$, where $V_{priv}$ is the set of all privacy concerning tokens in the data. From the inequality, it is obvious that the existing privacy notion is treating all tokens in the member sequence as private, leading to inflated membership scores in evaluation. In

this process, the true information leakage can be diluted or overshadowed on the sequence level (Figure 8), especially in long texts and documents. This masks the signals from the truly private tokens and leads to inaccurate auditing results (since we are evaluating the upper bounds). Moreover, a sequence-based analysis framework also fails to pinpoint the source of true leakage. This affects downstream tasks like unlearning: we cannot make the model forget the sensitive information, but rather entire documents that may contain useful general semantic knowledge.

With a token-level framework, users can compute leakage via every token completion. They can then easily visualize what tokens are memorized outputs and check if they are sensitive (Figure 1b). For auditors who know where the personally identifiable information (PII) is, they can also choose to directly check the model's memorization extent on the corresponding tokens. We build this interface and will explain in detail in Section 5. We want to highlight that we assume that users of the interface know what sensitive tokens are. This interface is not for automatically quantifying the privacy risk of a system on a data distribution, but rather an inspection tool for knowledgeable users, such as data owners and privacy auditors, to diagnose fine-grained information leakage. More importantly, this flexible framework can be adapted to audit $n$-gram privacy Duan et al. (2024); Liu et al. (2025), by summing up the token-based membership scores in any chosen $n$-gram as its memorization score.

## 4.2 TOKEN-LEVEL INFoRMIA

Our token-level framework relies on an MIA that can operate on the token level. We propose to conduct InfoRMIA (Equation 5) at each token generation step, treating all tokens $x$ as labels for their respective prefixal substrings, and then compute a token-based score. Additionally, we no longer have to curate a separate population dataset $Z$. Instead, we treat all possible tokens in the vocabulary other than the ground-truth $x$ as $z$. For example, if the ground-truth is 3, then $Z = \{z : z \in V \land z \neq 3\}$, where $V$ is the vocabulary of the tokenizer. The pseudocode can be found in Appendix C. In this way, we have a data-dependent $Z$ that removes the high cost of curating and computing on an independent population dataset. This makes the attack more feasible, especially for pretrained LLMs.

We want to emphasize that since $p(x|\theta) + \sum_{z \in Z} p(z|\theta) = \sum_{z \in V} p(z|\theta) = 1$ and $\sum_{z \in V} p(z) = \sum_{z \in V} \text{Avg}_{\theta_{\text{ref}}} p(z|\theta_{\text{ref}}) = \text{Avg}_{\theta_{\text{ref}}} \sum_{z \in V} p(z|\theta_{\text{ref}}) = 1$, we can have an equivalent formulation of the test statistic in Eqn 4 that does not require normalization (full derivation in Equation 10):

$$\text{Test Statistic} = \sum_{z \in Z} p(z) \log \left( \frac{p(\theta|x)}{p(\theta|z)} \right) \tag{6}$$

$$= \sum_{z \in Z} p(z) \log \left( \frac{p(x|\theta)p(z)}{p(z|\theta)p(x)} \right) + p(x) \log \left( \frac{p(x|\theta)p(x)}{p(x|\theta)p(x)} \right) \tag{7}$$

$$= \sum_{z \in V} p(z) \log \left( \frac{p(x|\theta)}{p(x)} \right) + \sum_{z \in V} p(z) \log \left( \frac{p(z)}{p(z|\theta)} \right) \tag{8}$$

$$= \log \left( \frac{p(x|\theta)}{p(x)} \right) + D_{\text{KL}} \left( p(z) \,||\, p(z|\theta) \right) \tag{9}$$

Equation 9 serves as an alternative form of our test statistic in Equation 5 that works with already normalized probabilities, where we can include $x$ in our $Z$ and compute the KL divergence on all possible token choices, without removing the ground truth token and gathering the remaining logits. But in our implementation, we reuse the equivalent form in the second last line of Equation 4.

## 4.3 FROM TOKEN-LEVEL TO SEQUENCE-LEVEL MIAS

The reigning privacy auditing and evaluation framework is on the sequence level. And for attackers with no knowledge of what private tokens are, the sequence-level notion is still the only choice. Here, we describe how to use token-level MIAs to perform sequence-level MIAs. We do not claim that this is the optimal way to evaluate token-level MIAs; this is more like a **proof of concept**. But our results in Section 6.3 prove that token-level MIA is useful, powerful and versatile. To obtain a sequence-based membership score, which is an aggregated notion as we argued, we inevitably need to aggregate token-based scores. For each given sequence $\mathbf{x} = \{x_1 x_2 \ldots x_k\}$, our token-

**Seq 491** (sample_index=81892, avg=0.331, avg_priv=nan)

Business Plan de e-commerce **Introduction** Le commerce électronique est en constante évolution, et pour réussir dans ce marché dynamique, il est essentiel d'avoir une stratégie solide et des objectifs clairs. Notre business plan pour notre entreprise e-commerce vise à définir nos actions et nos objectifs pour prospérer dans le secteur du commerce en ligne. **Stratégies clés** 1. **Segmentation du Marché**: Nous utiliserons les informations de nos clients pour diviser le marché en segments spécif

**Seq 434** (sample_index=20020, avg=0.330, avg_priv=nan)

Team Collaboration Platforms for Enhanced Pediatric Care Dear Team, In our continuous efforts to improve pediatric care services, we are excited to introduce a new team collaboration platform that will streamline our communication and enhance patient care outcomes. This platform aims to leverage technology to ensure efficient coordination among healthcare professionals and enhance the overall quality of care provided to our young patients. Key Features of

Figure 2: Two of the ten most memorized sequences by a finetuned GPT-2 model on ai4privacy dataset, identified by InfoRMIA, contain no private tokens. These pose little privacy risk, yet sequence-based frameworks overestimate their risk. See also Figure 7.

based MIA produces $k - 1$ membership scores $\{s_1, \ldots, s_{k-1}\}$, and the sequence-based score is Aggregate$(s_1, \ldots, s_{k-1})$.

The simplest aggregation is averaging. A stronger aggregation could depend on the model or underlying data distribution. Such tailored aggregation typically needs to be optimized on additional holdout data. However, such a model and dataset specific aggregator that requires additional knowledge and computation power is not always realistic. For practicality, we only evaluate generic aggregation methods, such as averaging and min-$k$, in this paper.

## 5 Token-Level Privacy Assessment Interface

In this section, we first demonstrate that our token-level framework, using InfoRMIA as the MIA backbone, can be used to visualize memorization on the token level. With knowledge of the sensitive tokens, users can conduct more insightful analyses of the privacy risks of the target model for given sequences, which can reveal much more than AUCs. Otherwise, auditing according to the average (sequence-level) privacy notion is recommended (See Section 6).

### 5.1 Visualizing Information Leakage on the Token Level

With token-based membership scores, we build a simple HTML interface to visualize a heatmap of token-level memorization over input text (Figures 1b, 2, 7, 8), where the darkness of the highlight reflects the degree of memorization. This fine-grained view enables more accurate privacy assessment, as auditors can directly inspect leakage on the actual private tokens.

We find empirical evidence supporting our intuition that sequence-level signals may not correspond to true privacy risks. Specifically, we observe a low correlation between sequence-level and private-token scores (Figure 1a) and discover that many of the most "memorized" sequences either contain disproportionately little private information (Figure 7) or no private information at all (Figure 2). Conversely, we also find evidence that signals from private tokens are often diluted by the presence of many common tokens in long texts (Figure 8).

This fine-grained analysis is only possible with our token-level framework and highlights the limitations of existing sequence-based notions of privacy. We believe this tool can be highly valuable for practitioners and auditors who need precise, interpretable privacy quantification.

### 5.2 Token-Level Analysis Reveals More Than AUCs

Our token-level framework also reveals insights that aggregate metrics like AUC cannot capture. For AG News (Appendix F.1), we hypothesize that sensitive information typically appears in personally identifiable information (PII). We therefore use SpaCy (Honnibal & Montani, 2017) to classify entities. Overall, token-level scores roughly follow a normal distribution (Figure 5). Tokens labeled `PERSON` and `WORK_OF_ART` have the highest average membership scores (Figure 3, Table 18), indicating that names of people and artworks are more likely to be memorized. Examining the top 1% of the highest-scoring tokens, we find that these two types of tokens also have the highest memorization rate (Table 18), reinforcing that PII is disproportionately more memorized.

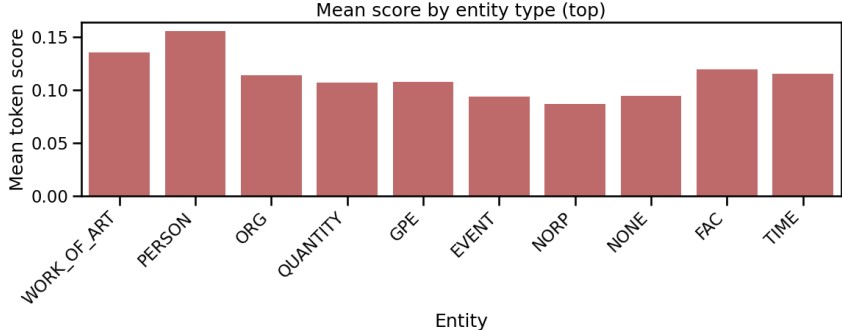

Figure 3: Average token-level membership scores across the most frequent named entity types on the AG News dataset. Tokens associated with names exhibit higher memorization signals, suggesting heterogeneous memorization behavior across semantic token types. The "None" category corresponds to tokens not identified as nouns. Details of the target model are provided in Section 6.1.

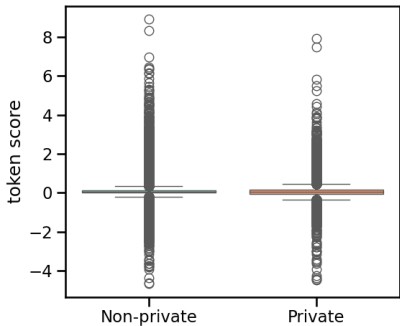

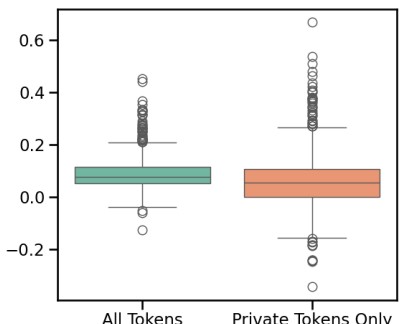

(a) Non-private tokens exhibit larger variance and higher average token-level membership scores, obscuring signals from private tokens.

(b) Aggregating token-level scores into sequence-level reduces private token membership signals despite higher token-level maxima.

Figure 4: Token-level and sequence-level membership score distributions for a fine-tuned GPT-2 model on the ai4privacy dataset. The results illustrate how sequence-level aggregation can obscure privacy leakage signals that originate from a small subset of private tokens. Additional details are provided in Table 19 and Figure 6.

For ai4privacy (Appendix F.2), each sequence includes a "privacy mask" that marks synthetic personal information. We can thus divide tokens into private and non-private sets. We find that the average membership score of private tokens is slightly lower than that of non-private tokens (Table 19, Figure 4a). This suggests that the high AUCs reported by sequence-level MIAs may largely reflect memorization of non-private content, which is less relevant for privacy. Hence, AUC alone is a poor indicator of true privacy risk in LLMs.

## 6 EXPERIMENTS

In this section, we present the attack performances of InfoRMIA and token-level InfoRMIA. We will describe the setup in Section 6.1, then show that InfoRMIA dominates RMIA and LiRA in Section 6.2. We then demonstrate our token-level framework's competitive performance in auditing sequence-level privacy (described in Section 4.3) on fintuned and pretrained LLMs in Section 6.3.

### 6.1 EXPERIMENTAL SETUP

**Setup** For easy benchmarking, we use the new ML Privacy Meter[2], which is an open-source Python library that audits privacy based on RMIA, released by the same lab behind the RMIA paper[3].

---

[2]https://github.com/privacytrustlab/ml_privacy_meter. InfoRMIA will be merged into the main branch via a pull request to replace RMIA as the core MIA engine.

[3]There are incorrect RMIA implementations online. For correctness, we opt for the library from the same lab/authors. See also Appendix E.3.

We evaluate on all three default benchmarks in the Privacy Meter: Purchase100 (Shokri et al., 2017), CIFAR-10 (Krizhevsky et al., 2009) and AG News (Zhang et al., 2015), which cover tabular, image and text datasets. Note that the AG News dataset is used for text generation instead of classification. The details of how we used the tool are in Appendix E. For those who are unfamiliar, the Privacy Meter takes in a configuration file that specifies meta information and hyperparameters. It will then train a set of models of the given architecture for the specified epochs, each on a randomly selected half of the dataset's training split, identical to LiRA and RIMA. The target model will be chosen randomly from the set, rendering the rest reference models. We also manually split the dataset and train the models in the same way when experimenting on ai4privacy in Table 2. We then conduct *offline* attacks on randomly sampled sets of target models' members and non-members, computing their membership scores under different attacks and evaluating the AUCs and TPR at small FPR levels.

**Model architectures**  We use GPT-2s (Radford et al., 2019) for AG News and ai4privacy, WideResNets-28-2 (Zagoruyko & Komodakis, 2016) for CIFAR10, and two two-layer MLPs for Purchase100. We use 1 epoch on AG News, and 4 epochs on the other two for Table 1, and 4 epochs on bothe datasets for Table 2. The other training details are in Appendix E.

## 6.2 InfoRMIA Dominates the Original RMIA

In both Tables 1 and 2, InfoRMIA dominates RMIA and LiRA in all cases. Besides higher AUCs, InfoRMIA greatly improves the TPR at very small FPR levels, indicating stronger member identification power without making (many) mistakes. As Carlini et al. (2022) argued, this metric is a better indicator of the true membership inference power. More importantly, we notice that InfoRMIA is less sensitive[4] to the size of the population data $Z$. This supports our theoretical derivation earlier, and is extremely valuable in practice, as the attack can now be run more accurately without a large pool of population data, reducing the computation overhead and real-life infeasibility.

Table 1: Comparison of AUC and TPR@0.1%FPR between RMIA, LiRA, and InfoRMIA, both with 4 reference models under different datasets and population sizes. For CIFAR-10, the Privacy Meter does not include augmentations in training and attacking, hence the lower numbers compared to the RMIA paper. For Purchase100, the Privacy Meter uses a much larger training set, a better evaluation per Suri et al. (2024). Since LiRA does not use $Z$, we replicate its numbers across different $|Z|$.

|  |  | **AG News** | | **CIFAR-10** | | **Purchase100** | |
|---|---|---|---|---|---|---|---|
| $|Z|$ |  | 100 | 1000 | 1000 | 10000 | 1000 | 10000 |
| **RMIA** | AUC | 0.8574 | 0.8766 | 0.8229 | 0.8327 | 0.5311 | 0.5432 |
|  | TPR@0.1%FPR | 0.00% | 1.60% | 0.00% | 0.00% | 0.00% | 0.00% |
| **InfoRMIA** | AUC | **0.8784** | **0.8784** | **0.8330** | **0.8330** | **0.5754** | **0.5754** |
|  | TPR@0.1%FPR | **12.0%** | **12.0%** | **5.82%** | **5.82%** | **0.32%** | **0.32%** |
| **LiRA** | AUC | 0.8641 | 0.8641 | 0.8242 | 0.8242 | 0.5398 | 0.5398 |
|  | TPR@0.1%FPR | 1.80% | 1.80% | 0.12% | 0.12% | 0.12% | 0.12% |

## 6.3 Solving Sequence-Level Membership Inference with Token-Level Membership Signals

**Finetuned LLMs**  We also demonstrate that token-level membership inference can be used to conduct sequence-based membership inference with competitive performance, despite not being designed for it. We show in Table 2 that token-level membership inference with simple averaging yields competitive performance in sequence-level membership inference evaluation.

---

[4]We audit the privacy risk of individual target models in this table. This makes the second term in InfoRMIA's test statistic (Equation 4) identical for all test queries $x$'s, keeping the order of test statistics among test queries, and ultimately the attack performance, unchanged by the varying $|Z|$.

Table 2: Comparison of AUC and TPR@1%FPR between the sequence-based and token-based InfoRMIA, and the original RMIA and LiRA when attacking finetuned LLMs. The epochs column refers to the finetuning epochs. We use one reference model throughout the evaluation.

| Datasets | Epochs | RMIA | | InfoRMIA | | InfoRMIA (token) | | LiRA | |
|---|---|---|---|---|---|---|---|---|---|
| | | AUC | TRP@FPR | AUC | TRP@FPR | AUC | TRP@FPR | AUC | TRP@FPR |
| AG News | 1 | 0.839 | 0.00% | **0.843** | **23.0%** | 0.836 | 20.2% | 0.795 | 3.6% |
| | 4 | **0.945** | 0.00% | **0.945** | 16.2% | 0.942 | **20.6%** | 0.882 | 9.00% |
| ai4privacy | 1 | 0.643 | 6.6% | **0.644** | **10.6%** | 0.620 | 9.0% | 0.630 | 3.8% |
| | 4 | 0.821 | 26.0% | **0.822** | **27.2%** | 0.804 | 23.2% | 0.782 | 10.4% |

**Pretrained LLMs** Besides finetuned models, we also evaluate our newly proposed method on pretrained models (Table 3-8) on the most popular benchmark, MIMIR (Duan et al., 2024). As Duan et al. (2024) pointed out, reference-based MIAs do not perform well on MIMIR due to the lack of quality reference models. Ideally, the reference models should have the same model architecture and be trained on the same data distribution as the target LLM, but with (partially) disjoint datasets (which effectively makes them OUT models in RMIA's terminology). However, for pretrained LLMs, it is not possible to find such perfect reference models. In our experiments, we find that using an earlier snapshot of the LLM is more useful, so we use the step-1 `Pythia-160M` checkpoint as our sole reference model[5]. This is a very practical solution, as this checkpoint can be easily trained with lower-end hardware within a short period of time, even if it is not available. Because it is very OUT, it gives better results than using later checkpoints as the reference models (See Appendix G.3). Since no in-distributional population data can be easily obtained, we only evaluate the **token-level InfoRMIA** in our experiments, which has a similar computational complexity as the *Ref* method (Carlini et al., 2021) in MIMIR. The explanation of each method in the table is in Appendix F.3.1.

We observe that our token-level InfoRMIA, although not specifically designed for sequence-level membership inference, achieves one of the strongest membership inference performances (Table 3, 4) even when using only a single, less ideal reference model. In Tables 6–8, we report results obtained by using the step-1 checkpoint of the target model as the reference and observe minimal change in utility. We further show that our method is the strongest reference-based MIA for pretrained LLMs, outperforming prior reference-model approaches (Ref). We also find that simple averaging outperforms the min-$k$ aggregation when targeting high true-positive rates at low false-positive rates (TPR at small FPR). This is expected, as min-$k$ aggregation essentially acts as a non-member detector: non-members tend to contain more low-probability tokens (Shi et al., 2024). Consequently, min-$k$ is less suitable for high-precision member detection with minimal errors. However, when evaluating using AUC (Table 5), the ordering reverses. This aligns with the argument by Carlini et al. (2022) that AUC can be misleading as a privacy metric. Nonetheless, many recent works evaluating on MIMIR still report only AUC in their main text, which may encourage a suboptimal trend for future development of LLM MIAs.

## 7 CONCLUSION

In this paper, we propose a new information-theoretic formulation of the membership inference test. The resulting attack, InfoRMIA, consistently outperforms the prior state-of-the-art RMIA across tabular, image, and text datasets, while eliminating RMIA's reliance on a large population set. Its superior performance stems from a more principled statistical test and the use of a continuous score rather than a discrete one.

We then introduce a new perspective for analyzing privacy risks in LLMs through a token-level analysis framework. It can reveal which tokens are memorized within each sequence, while achieving higher membership inference power. By uncovering fine-grained memorization patterns, our token-level framework enables more precise privacy risk estimation, and opens the door to downstream applications such as targeted machine unlearning and token-guided data reconstruction and extraction. We leave the systematic exploration of these applications to future work.

---

[5]We use the non-deduped version as it sees fewer unique training sequences and hence more OUT.

REPRODUCIBILITY STATEMENT

We have described the training details, including model architectures, dataset splits, and essential hyperparameters in the paper. We will release the code and submit pull requests to the benchmarks (ML-Privacy-Meter and MIMIR) to facilitate the reproduction of the results presented in the paper.

DISCLOSURE ON THE USE OF LLMs

LLMs have only been used in polishing writing and non creative part of coding, such as refactoring, annotating the code, and standardizing the format.

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

## A  THE MEMBERSHIP INFERENCE GAME

Membership inference is often formulated as an inference game. According to the auditing modes (privacy of a fixed model/data record/training algorithm), the game formulation also varies (Ye et al., 2022). Here, we provide the game formulation when auditing the privacy of a fixed model.

**Definition 1** *(**Membership Inference Game** (Yeom et al., 2018; Ye et al., 2022; Carlini et al., 2022; Zarifzadeh et al., 2024; Tao & Shokri, 2025)) Let $\pi$ be the data distribution, and let $\mathcal{T}$ be the training algorithm.*

1. *The* challenger *samples a training dataset $D \longleftarrow \pi$, and trains a model $\theta \longleftarrow \mathcal{T}(D)$.*

2. *The* challenger *samples a data record $z_0 \longleftarrow \pi$ from the data distribution, and a training data record $z_1 \longleftarrow D$.*

3. *The* challenger *flips a fair coin to get the bit $b \in \{0, 1\}$, and sends the target model $\theta$ and data record $z_b$ to the* adversary.

4. *The* adversary *gets access to the data distribution $\pi$ and access to the target model, and outputs a bit $\hat{b} \longleftarrow \mathcal{A}(\theta, z_b)$.*

5. *If $\hat{b} = b$, output 1 (success). Otherwise, output 0.*

## B    DERIVATION OF INFORMIA SCORES

$$
\begin{aligned}
\text{Test Statistic} &= \sum_{z \in Z} p(z) \log\left(\frac{p(\theta|x)}{p(\theta|z)}\right) \\
&= \sum_{z \in Z} p(z) \log\left(\frac{p(x|\theta)p(z)}{p(z|\theta)p(x)}\right) \\
&= \sum_{z \in Z} p(z) \log\left(\frac{p(x|\theta)p(z)}{p(z|\theta)p(x)}\right) + p(x) \log\left(\frac{p(x|\theta)p(x)}{p(x|\theta)p(x)}\right) \\
&= \sum_{z \in V} p(z) \log\left(\frac{p(x|\theta)p(z)}{p(z|\theta)p(x)}\right) \\
&= \sum_{z \in V} p(z) \log\left(\frac{p(x|\theta)}{p(x)}\right) + \sum_{z \in V} p(z) \log\left(\frac{p(z)}{p(z|\theta)}\right) \\
&= \log\left(\frac{p(x|\theta)}{p(x)}\right) + D_{\text{KL}}\left(p(z) \,||\, p(z|\theta)\right)
\end{aligned}
\tag{10}
$$

## C    PSEUDOCODE OF RMIA AND INFORMIA

In this section, we outline the consolidated pseudocode (Alg 1) of RMIA, InfoRMIA, and token-level InfoRMIA for an easy comparison of the attacks. Lines 5 to 9 clearly show that the token-level InfoRMIA has the smallest computation overhead due to the absence of the additional population dataset. Its $p(z|\theta)$ terms are also directly obtainable from the calculation of $P(x|\theta)$ since each prediction step computes the softmax scores over the entire vocabulary. Note that for LLMs, the hyperparameter $a$ is optimal at $a = 1$ (Hayes et al., 2025a). When using token-level InfoRMIA, each $x$ is a prefixal subsequence instead of a full sequence (See Section 4.1).

## D    DEFENDING INFORMIA

Similar to all other MIA techniques, InfoRMIA relies on a model's memorization for successful attacks. It does not use additional information or require higher levels of access to the target models. Therefore, differentially private training algorithms that reduce memorization of training data are still effective in defending against InfoRMIA.

## E    IMPLEMENTATION DETAILS

### E.1    RMIA REFERENCE MODEL TRAINING

We use the default hyperparameters in ML Privacy Meter to train target and reference models when comparing the original and InfoRMIA, except for the number of epochs. For each dataset, the hyperparameter choices can be found in `https://github.com/privacytrustlab/ml_privacy_meter/tree/master/configs`. For CIFAR-10 and Purchase-100, we use 100 epochs, while for AG News, we use 1 epoch. We use the one-liner command `python run_mia.py --cf configs/xxx.yaml` to run all the experiments, where the yaml files correspond to the respective default configs in the GitHub repo.

For ai4privacy experiments, we train target and reference models, which are initiated from GPT-2 models, on randomly selected halves of the training set, identical to the setup of LiRA and RMIA.

### E.2    SOFTWARE AND HARDWARE

For all transformer models and language datasets, we use the libraries from Huggingface. The training process also uses Huggingface's Trainer class. All computations are done on two NVIDIA RTX-3090 and two H100 GPUs.

---

**Algorithm 1** MIA Score Computation with Offline **RMIA**, **InfoRMIA** or **Token-Level InfoRMIA**, modified from Zarifzadeh et al. (2024).

---

**Input:** Target model $\theta$, a set of $k$ reference models $\Theta$, target query $x$, hyperparameters $\gamma, a$, population dataset $Z$ (only for RMIA and InfoRMIA)

**Output:** Membership score $\text{Score}_{\text{MIA}}(x; \theta)$ of $x$ given the target model $\theta$

1: Compute $p(x|\theta)$ and $p(x|\theta_r)$ for all $\theta_r \in \Theta$

2: $p(x)_{\text{OUT}} \leftarrow \frac{1}{k} \sum_{\theta_r \in \Theta} p(x|\theta_r)$

3: $p(x) \leftarrow \frac{1}{2} \left( (1+a)p(x)_{\text{OUT}} + (1-a) \right)$

4: $\text{Ratio}_x \leftarrow \dfrac{p(x|\theta)}{p(x)}$

5: **if** Token-Level InfoRMIA **then**

6:      Take all other tokens except the ground-truth as $z$             ▷ See Section 4.2

7: **else**

8:      Take population data from the population dataset $Z$ as $z$

9: **end if**

10: **for** each $z$ **do**

11:      Compute $p(z|\theta)$ and $p(z|\theta_r)$ for all $\theta_r \in \Theta$

12:      $p(z) \leftarrow \frac{1}{k} \sum_{\theta_r \in \Theta} \text{Pr}(z|\theta_r)$

13:      $\text{Ratio}_z \leftarrow \dfrac{p(z|\theta)}{p(z)}$

14: **end for**

15: **if** RMIA **then**

16:      $\text{Score}_{\text{MIA}}(x; \theta) \leftarrow \frac{1}{|Z|} \sum_{z \in Z} \mathbb{I}\left( \dfrac{\text{Ratio}_x}{\text{Ratio}_z} \geq \gamma \right)$

17: **else**                                 ▷ InfoRMIA and Token-Level InfoRMIA

18:      $\text{Score}_{\text{MIA}}(x; \theta) \leftarrow \log \text{Ratio}_x - \sum_z p(z) \log \text{Ratio}_z$           ▷ See Equation 5

19: **end if**

---

### E.3 INFORMIA SIGNAL COMPUTATION

Several public implementations of RMIA differ in subtle but important details, which can lead to inconsistent results in practice. Since the core computation of InfoRMIA closely resembles RMIA, implementation errors can easily propagate. To facilitate reproducibility, we provide a reference implementation of the InfoRMIA signal computation in Python below. The official implementation is also available in the latest release of PRIVACY METER.

Listing 1: Reference implementation of the InfoRMIA test statistic computation.

```python
def run_informia(
    target_model_idx: int,
    all_signals: np.ndarray,
    population_signals: np.ndarray,
    all_memberships: np.ndarray,
    num_reference_models: int,
    offline_a: float,
) -> np.ndarray:
    """
    Compute InfoRMIA membership scores.

    Args:
        target_model_idx: Index of the target model.
        all_signals: Softmax scores of all samples.
        population_signals: Softmax scores of population samples.
        all_memberships: Membership matrix for all models.
        num_reference_models: Number of reference models.
        offline_a: Offline correction coefficient used
            to approximate p(x) using P_out.
```

```
    Returns:
        Membership inference scores for all samples.
        Larger values indicate higher membership likelihood.
    """

    # Target model signals
    target_signals = all_signals[:, target_model_idx]

    out_signals = get_rmia_out_signals(
        all_signals,
        all_memberships,
        target_model_idx,
        num_reference_models,
    )

    mean_out_x = np.mean(out_signals, axis=1) # P_out(x)
    mean_x = (
        ((1 + offline_a) / 2) * mean_out_x
        + ((1 - offline_a) / 2)
    ) # Offline estimation of P(x) according to RMIA

    # log (p(x|theta) / p(x))
    log_ratio_x = np.log(target_signals.ravel() / mean_x)

    population_memberships = np.zeros_like(
        population_signals,
        dtype=bool,
    )  # population samples are OUT

    z_signals = population_signals[:, target_model_idx]
    z_out_signals = get_rmia_out_signals(
        population_signals,
        population_memberships,
        target_model_idx,
        num_reference_models,
    )

    mean_out_z = np.mean(z_out_signals, axis=1)
    mean_z = (
        ((1 + offline_a) / 2) * mean_out_z
        + ((1 - offline_a) / 2)
    )

    prob_ratio_z = z_signals.ravel() / mean_z

    test_statistic = (
        log_ratio_x
        - np.sum(mean_z * np.log(prob_ratio_z))
        / mean_z.sum()
    )

    return test_statistic
```

When computing log-likelihood terms, we recommend using the smallest `epsilon` value that avoids numerical instability. Extremely small softmax probabilities may otherwise cause the logarithm to be dominated by the clipping constant. For clarity, no `epsilon` is applied in the reference implementation shown above.

### E.4 OTHER DETAILS

When computing the InfoRMIA test statistics, we use the second last line in Equation 4 because it is easier and more similar to the computation of RMIA's test statistics.

# F   DATASETS

## F.1   AG NEWS

AG News (Zhang et al., 2015) is a news dataset that contains four categories of news articles. Its training set size is 120,000. In our experiment, we ignore the labels column and train autoregressive models on the text only, identical to Mattern et al. (2023) and Tao & Shokri (2025). In implementation, we use the `datasets` pacakge from huggingface to load the dataset. The max sequence length is set to 512. The preprocessing and loading script is adapted from the Privacy Meter at `https://github.com/privacytrustlab/ml_privacy_meter/blob/e384af8fd9319b8eeb1303aa82474df1441e3c59/dataset/agnews.py`.

## F.2   AI4PRIVACY

The ai4privacy dataset we used is the pii-masking-300k variant, that can be access at `https://huggingface.co/datasets/ai4privacy/pii-masking-300k`. It is divided into two parts: OpenPII-220k and FinPII-80k. The FinPII has additional classes that are specific to the Finance and Insurance domains. In this dataset, there is a "privacy_mask" column that marks the beginning and end location for each piece of private information. Thus, we can use this information to categorize each token as private or non-private. It also assigns a type to private substrings, such as last names or email addresses, enabling us to do more interesting analysis.

## F.3   THE MIMIR BENCHMARK

MIMIR is a benchmark based on the Pile (Gao et al., 2020) dataset, where non-members highly overlapped with any member sequence are removed from the evaluation. Since the members and non-members are randomly shuffled before being split, MIMIR avoids the error of having a large distributional shift between the two sets (Maini et al., 2024; Das et al., 2025). It is also one of the most active benchmarks with official implementations of recent methods such as the MinK++ (Zhang et al., 2025), ReCaLL (Xie et al., 2024), and DC-PDD (Zhang et al., 2024).

### F.3.1   EXPLANATIONS OF ALL ATTACK METHODS

We will briefly explain the score formulation of each attacking method in the MIMIR benchmark. The full details of each attack can be found at the respective papers. Note that in this benchmark, the higher the score is, the less likely it is a member.

- **LOSS** (Yeom et al., 2018): the average loss values of the sequence
- **Zlib** (Carlini et al., 2021): the calibrated loss values by the entropy, estimated by the length after zlib compression
- **Min-K%** (Shi et al., 2024): the average of the bottom-$k\%$ of token probabilities, and taking the negative (to align the score's sign with MIMIR's standard)
- **Min-K%++** Zhang et al. (2025): the average of the bottom-$k\%$ of token probabilities calibrated by the mean and variance of each token position's softmax output distributions, and taking the negative
- **DC-PDD** (Zhang et al., 2024): the average token probabilities calibrated by token frequencies calculated on a reference dataset, and take the negative
- **Ref** (Carlini et al., 2021): the average token loss gap between a reference model and the target model

### F.3.2   WHY RECALL WAS EXCLUDED IN OUR TABLE

The ReCaLL attack pushed to the MIMIR benchmark is a simplified version (according to the author) and very problematic. There is an implicit information leakage about the membership labels in crafting the attack signals, making the result unreliable and unfair. By right, the attacker should not be able to distinguish any non-member from any member, which means the attacker has no information that can be used to tell non-members apart from members before the attack. However, the

ReCaLL attack in MIMIR explicitly uses non-members in the evaluation set as its "non-member" prefix to be prepended to all sequences, making the attacker aware of the membership label of certain non-members. Although the label information is not explicitly used in the attack, it is an implicit information leak that can be used to tell apart the two sets. Hence, the current implementation of the ReCaLL attack in MIMIR is incorrect. We will include it once the official implementation is fixed.

# G ADDITIONAL RESULTS

## G.1 MIMIR RESULTS

We provide more results on the MIMIR benchmark here. In particular, we use the `ngram<0.8` split. Table 3 and 4 show the results on TPR @1% FPR and TPR@0.1%FPR respectively, while Table 5 shows the AUCs. The results show that our method has strongest inference power (highest TPR at small FPRs), while achieving very competitive results on AUCs. It is also stronger than the prior reference-based MIA, Ref (Carlini et al., 2021).

Table 3: TPR @1% FPR on MIMIR with deduped Pythia models. on MIMIR benchmark with deduped Pythia models. The *Neighbor* method is not included due to its computational complexity and relatively inferior performance reported in prior works. *ReCaLL* is not included for reasons in Appendix F.3.2. *Ref* method is evaluated using the checkpoint of Pythia-160m after the first step as the reference model. Our method (*InfoRMIA*) is the token-based InfoRMIA that does not require additional population data. InfoRMIA1 uses averaging to aggregate, while InfoRMIA2 uses min-k%, using the same hyperparameter $k$ as *Min-K%* and *Min-K%++*. Bold numbers are the best, and the underlined are the best reference-based.

| | Wikipedia | | | | Github | | | | Pile CC | | | | PubMed Central | | | |
|---|---|---|---|---|---|---|---|---|---|---|---|---|---|---|---|---|
| **Method** | 160M | 1.4B | 2.8B | 6.9B | 160M | 1.4B | 2.8B | 6.9B | 160M | 1.4B | 2.8B | 6.9B | 160M | 1.4B | 2.8B | 6.9B |
| Loss | 0.9 | 0.6 | 0.6 | 0.6 | 13.1 | 13.3 | 21.9 | 13.2 | 0.4 | 0.7 | 0.8 | 0.9 | 0.7 | 0.4 | 0.6 | 0.4 |
| Zlib | 1.3 | 0.7 | 0.8 | 0.6 | 14.3 | 16.9 | **24.0** | **15.5** | 0.7 | 0.7 | 0.9 | **1.5** | 0.3 | 0.4 | 0.5 | 0.4 |
| Min-K% | **1.4** | 0.9 | 0.6 | 0.5 | 12.0 | 13.1 | 21.8 | 13.0 | 0.5 | 0.6 | 0.7 | 1.0 | 0.6 | 0.2 | 0.6 | 0.4 |
| Min-K%++ | 1.2 | 0.7 | 0.6 | 1.0 | 11.2 | 12.8 | 18.1 | 12.8 | 1.1 | **1.1** | **1.2** | **1.5** | 0.6 | 0.4 | 0.5 | 0.6 |
| DC-PDD | 0.9 | 0.4 | **1.2** | **1.4** | 10.8 | 11.3 | 9.8 | 10.7 | 0.4 | **1.1** | 0.6 | 1.1 | **1.5** | 0.8 | **1.3** | 1.3 |
| Ref | 0.9 | 0.8 | 0.7 | 0.6 | 13.4 | 13.9 | 20.3 | 14.7 | 0.6 | 0.7 | 0.8 | 1.0 | 0.8 | 0.6 | 0.5 | 0.4 |
| InfoRMIA1 | 0.8 | 0.9 | 0.6 | 0.9 | **14.7** | **17.7** | 21.2 | **15.5** | **1.2** | 0.9 | 0.7 | 0.6 | 1.0 | 0.5 | 0.3 | 0.4 |
| InfoRMIA2 | 1.1 | **1.2** | 0.9 | 0.7 | 13.0 | 14.1 | 18.3 | 14.5 | 0.4 | 0.5 | 0.7 | 0.5 | 1.2 | **1.4** | **1.3** | **1.6** |

| | ArXiv | | | | DM Mathematics | | | | HackerNews | | | | Average | | | |
|---|---|---|---|---|---|---|---|---|---|---|---|---|---|---|---|---|
| **Method** | 160M | 1.4B | 2.8B | 6.9B | 160M | 1.4B | 2.8B | 6.9B | 160M | 1.4B | 2.8B | 6.9B | 160M | 1.4B | 2.8B | 6.9B |
| Loss | 0.7 | 0.7 | 0.4 | 0.8 | 0.5 | 0.5 | 1.1 | **1.1** | 0.9 | 0.7 | 0.6 | 0.8 | 2.5 | 2.4 | 3.7 | 2.5 |
| Zlib | 0.5 | 0.2 | 0.4 | 0.7 | 1.1 | 0.9 | 0.9 | 0.6 | 0.6 | 1.0 | 1.0 | 1.0 | 2.7 | 3.0 | **4.1** | **2.9** |
| Min-K% | 0.3 | 0.3 | 0.4 | 0.7 | 0.8 | 0.6 | 0.2 | 0.4 | 0.7 | 0.9 | 0.7 | 1.1 | 2.3 | 2.4 | 3.6 | 2.4 |
| Min-K%++ | **1.1** | **1.9** | **1.2** | **1.4** | 1.0 | 1.0 | **1.2** | 1.0 | 0.7 | 0.5 | 1.1 | 0.7 | 2.4 | 2.6 | 3.4 | 2.7 |
| DC-PDD | 0.5 | 1.0 | 0.9 | 0.5 | 0.5 | 0.4 | 0.2 | 0.1 | 1.3 | 1.0 | 0.5 | 1.2 | 2.3 | 2.3 | 2.1 | 2.3 |
| Ref | 0.8 | 0.5 | 0.5 | 0.6 | **1.2** | 1.0 | **1.2** | 0.7 | 1.4 | 0.7 | 0.7 | 0.7 | 2.7 | 2.6 | 3.5 | 2.7 |
| InfoRMIA1 | 0.3 | 0.5 | 0.4 | 0.6 | 0.7 | **1.3** | 1.0 | **1.1** | 1.5 | **1.2** | **1.7** | 1.3 | **2.9** | **3.3** | 3.7 | **2.9** |
| InfoRMIA2 | 0.1 | 0.3 | 0.3 | 0.4 | **1.2** | 0.8 | 1.1 | 0.7 | **1.7** | **1.2** | 1.0 | **1.8** | 2.7 | 2.8 | 3.4 | **2.9** |

## G.2 MIMIR RESULTS WHEN USING THE FIRST STEP CHECKPOINT OF THE TARGET MODEL AS THE REFERENCE

Instead of using the first step checkpoint of `Pythia-160m`, we use the checkpoint corresponding to the target model, e.g., we use the `Pythia-1.4b:step1` as the reference when attacking `Pythia-1.4b-deduped`. We found in Table 6, 7 and 8 that the difference in utility is minimal. Therefore, it might be better to stick to snapshots of smaller models as reference models for cost-sensitive auditors. Moreover, using the first step checkpoint of the small LLM as the reference has

Table 4: TPR @0.1% FPR on MIMIR with deduped Pythia models.

| Method | Wikipedia | | | | Github | | | | Pile CC | | | | PubMed Central | | | |
|---|---|---|---|---|---|---|---|---|---|---|---|---|---|---|---|---|
| | 160M | 1.4B | 2.8B | 6.9B | 160M | 1.4B | 2.8B | 6.9B | 160M | 1.4B | 2.8B | 6.9B | 160M | 1.4B | 2.8B | 6.9B |
| Loss | 0.0 | 0.1 | **0.2** | 0.1 | 5.7 | 4.8 | 7.9 | 5.2 | 0.0 | **0.2** | 0.1 | 0.2 | **0.0** | 0.0 | 0.0 | 0.0 |
| Zlib | **0.1** | 0.1 | 0.1 | 0.1 | **8.4** | **5.7** | **8.6** | **6.9** | 0.0 | **0.2** | 0.2 | **0.3** | **0.0** | 0.0 | 0.0 | 0.0 |
| Min-K% | 0.0 | 0.1 | **0.2** | 0.1 | 5.7 | 4.8 | 8.0 | 4.9 | 0.0 | **0.2** | 0.1 | 0.2 | **0.0** | 0.0 | 0.0 | 0.0 |
| Min-K%++ | 0.0 | 0.0 | 0.1 | 0.0 | 6.1 | 3.5 | 4.6 | 2.1 | **0.1** | **0.2** | 0.1 | **0.3** | **0.0** | 0.0 | 0.0 | 0.0 |
| DC-PDD | 0.0 | 0.0 | 0.0 | 0.0 | 3.7 | 0.3 | 0.3 | 1.1 | 0.0 | 0.0 | **0.3** | 0.2 | **0.0** | 0.0 | 0.1 | 0.1 |
| Ref | 0.0 | 0.0 | 0.0 | 0.0 | 4.3 | _4.3_ | _2.2_ | _3.5_ | _0.1_ | **0.2** | **0.3** | **0.3** | **0.0** | _0.1_ | 0.0 | 0.0 |
| InfoRMIA1 | _0.1_ | **0.2** | **0.2** | **0.2** | 0.0 | 0.0 | 0.6 | 1.0 | _0.1_ | 0.1 | 0.1 | 0.1 | _0.0_ | _0.1_ | **0.3** | **0.3** |
| InfoRMIA2 | 0.0 | 0.0 | 0.1 | 0.1 | _4.8_ | 0.0 | 0.9 | 0.2 | _0.1_ | 0.1 | 0.1 | 0.2 | _0.0_ | 0.0 | 0.0 | 0.0 |

| Method | ArXiv | | | | DM Mathematics | | | | HackerNews | | | | Average | | | |
|---|---|---|---|---|---|---|---|---|---|---|---|---|---|---|---|---|
| | 160M | 1.4B | 2.8B | 6.9B | 160M | 1.4B | 2.8B | 6.9B | 160M | 1.4B | 2.8B | 6.9B | 160M | 1.4B | 2.8B | 6.9B |
| Loss | 0.1 | 0.0 | 0.0 | **0.1** | 0.0 | 0.0 | 0.2 | 0.0 | 0.1 | 0.0 | 0.0 | 0.0 | 0.8 | 0.7 | 1.2 | 0.8 |
| Zlib | 0.0 | 0.0 | 0.0 | 0.0 | 0.0 | 0.0 | 0.0 | 0.0 | 0.1 | 0.2 | 0.2 | 0.2 | **1.2** | **0.9** | **1.3** | **1.1** |
| Min-K% | 0.0 | 0.0 | 0.0 | 0.0 | 0.0 | 0.0 | 0.0 | 0.0 | 0.2 | 0.1 | 0.1 | 0.1 | 0.8 | 0.7 | 1.2 | 0.8 |
| Min-K%++ | 0.0 | 0.0 | 0.0 | 0.0 | 0.1 | 0.0 | **0.5** | 0.2 | 0.2 | 0.0 | 0.1 | 0.0 | 0.9 | 0.5 | 0.8 | 0.4 |
| DC-PDD | 0.0 | 0.0 | **0.2** | 0.0 | 0.0 | 0.0 | 0.0 | 0.0 | 0.0 | 0.2 | 0.1 | 0.0 | 0.5 | 0.1 | 0.1 | 0.2 |
| Ref | **0.2** | _0.1_ | 0.0 | 0.0 | **0.2** | 0.0 | 0.1 | 0.1 | _0.1_ | 0.0 | 0.0 | 0.2 | 0.7 | _0.7_ | _0.4_ | _0.6_ |
| InfoRMIA1 | 0.1 | 0.0 | _0.0_ | _0.0_ | 0.1 | **0.3** | **0.3** | **0.6** | 0.0 | 0.0 | 0.0 | 0.0 | 0.1 | 0.1 | 0.2 | 0.3 |
| InfoRMIA2 | 0.0 | 0.0 | _0.0_ | _0.0_ | 0.0 | 0.0 | 0.0 | 0.0 | _0.1_ | **0.3** | **0.3** | **0.3** | _0.7_ | 0.1 | 0.2 | 0.1 |

Table 5: AUC results on MIMIR benchmark with deduped Pythia models.

| Method | Wikipedia | | | | Github | | | | Pile CC | | | | PubMed Central | | | |
|---|---|---|---|---|---|---|---|---|---|---|---|---|---|---|---|---|
| | 160M | 1.4B | 2.8B | 6.9B | 160M | 1.4B | 2.8B | 6.9B | 160M | 1.4B | 2.8B | 6.9B | 160M | 1.4B | 2.8B | 6.9B |
| Loss | 50.2 | 51.0 | 51.7 | 51.6 | 63.7 | 65.8 | 71.2 | 67.6 | 49.5 | 50.1 | 50.1 | 51.3 | 49.9 | 49.8 | 49.9 | 50.5 |
| Zlib | **51.0** | 51.8 | 52.4 | 52.3 | **65.6** | **67.2** | **72.2** | 68.8 | 49.6 | 50.2 | 50.3 | 51.2 | 50.0 | 50.0 | 50.0 | 50.6 |
| Min-K% | 48.6 | 50.6 | 51.6 | 51.4 | 63.6 | 65.9 | 71.4 | 68.0 | 50.0 | 51.0 | 50.5 | 51.9 | 50.4 | 50.2 | 50.4 | 51.0 |
| Min-K%++ | 47.7 | **52.3** | **53.7** | **52.4** | 61.4 | 65.7 | 70.7 | **69.1** | 49.8 | 51.1 | 49.9 | 51.7 | 50.9 | 50.6 | **51.2** | **52.3** |
| DC-PDD | 49.0 | 50.6 | 52.4 | 51.8 | 64.9 | 66.2 | 71.4 | 69.0 | 49.6 | 51.1 | **51.2** | **51.9** | 50.5 | **51.0** | 50.6 | 51.1 |
| Ref | 50.0 | 50.8 | _51.6_ | _51.4_ | 63.9 | 66.0 | _71.4_ | _67.9_ | 49.4 | 50.0 | 50.0 | 51.2 | 49.8 | 49.7 | 49.8 | 50.4 |
| InfoRMIA1 | _50.9_ | _50.8_ | 51.0 | 51.2 | _65.0_ | _66.1_ | 70.8 | 67.0 | 49.4 | 49.6 | 49.8 | 50.5 | 50.2 | 49.7 | 49.5 | 49.8 |
| InfoRMIA2 | 50.0 | 50.3 | 51.1 | 51.1 | 63.5 | 65.3 | 70.6 | 66.9 | **50.6** | **51.1** | 50.8 | 51.7 | **51.4** | _50.4_ | 50.2 | 50.7 |

| Method | ArXiv | | | | DM Mathematics | | | | HackerNews | | | | Average | | | |
|---|---|---|---|---|---|---|---|---|---|---|---|---|---|---|---|---|
| | 160M | 1.4B | 2.8B | 6.9B | 160M | 1.4B | 2.8B | 6.9B | 160M | 1.4B | 2.8B | 6.9B | 160M | 1.4B | 2.8B | 6.9B |
| Loss | 50.7 | 51.4 | 51.9 | 52.5 | 49.0 | 48.6 | 48.3 | 48.4 | 49.2 | 50.4 | 51.2 | 51.7 | 51.8 | 52.4 | 53.5 | 53.4 |
| Zlib | 50.0 | 50.8 | 51.3 | 51.8 | 48.2 | 48.1 | 48.0 | 48.1 | 49.6 | 50.2 | 50.9 | 51.0 | 52.0 | 52.6 | 53.6 | 53.4 |
| Min-K% | 50.0 | 51.2 | 52.2 | 52.7 | 49.4 | 49.3 | 49.1 | 49.3 | 50.2 | 51.3 | 52.4 | 53.0 | 51.7 | 52.8 | 53.9 | 53.9 |
| Min-K%++ | 48.7 | 51.2 | **53.1** | 52.8 | **49.9** | **50.0** | 50.3 | 50.2 | 50.9 | 51.1 | 52.3 | 53.7 | 51.3 | **53.1** | 54.4 | **54.6** |
| DC-PDD | 50.4 | **52.0** | 52.9 | **52.9** | 49.0 | 49.3 | 49.8 | 49.7 | 50.7 | 51.8 | **53.0** | 53.9 | 52.0 | 53.1 | **54.5** | 54.3 |
| Ref | 50.3 | 51.0 | 51.5 | 52.1 | 48.8 | 48.5 | 48.3 | 48.3 | 49.1 | 50.4 | 51.2 | 51.7 | 51.6 | 52.3 | 53.4 | 53.3 |
| InfoRMIA1 | 50.3 | 51.1 | 51.1 | 51.5 | 48.0 | 47.6 | 47.9 | 47.9 | 50.4 | 50.7 | 51.0 | 51.3 | 52.0 | 52.2 | 53.0 | 52.7 |
| InfoRMIA2 | _50.8_ | _51.2_ | _51.6_ | _52.3_ | 49.0 | _49.1_ | _49.0_ | _48.9_ | _50.4_ | **51.9** | 52.4 | 53.1 | **52.2** | _52.7_ | _53.7_ | _53.5_ |

another benefit: if the checkpoint is not publicly available, training the small LLM for one step is computationally cheap. Normal users can obtain the checkpoint and run the attack with low end computes and short training period.

Table 6: TPR @1% FPR on MIMIR benchmark with deduped Pythia models when using the first step checkpoint of the target model as the reference.

| Method | Wikipedia | | | | Github | | | | Pile CC | | | | PubMed Central | | | |
|---|---|---|---|---|---|---|---|---|---|---|---|---|---|---|---|---|
| | 160M | 1.4B | 2.8B | 6.9B | 160M | 1.4B | 2.8B | 6.9B | 160M | 1.4B | 2.8B | 6.9B | 160M | 1.4B | 2.8B | 6.9B |
| Loss | 0.9 | 0.6 | 0.6 | 0.6 | 13.1 | 13.3 | 21.9 | 13.2 | 0.4 | 0.7 | 0.8 | 0.9 | 0.7 | 0.4 | 0.6 | 0.4 |
| Zlib | 1.3 | 0.7 | 0.8 | 0.6 | 14.3 | 16.9 | **24.0** | **15.5** | 0.7 | 0.7 | 0.9 | **1.5** | 0.3 | 0.4 | 0.5 | 0.4 |
| Min-K% | **1.4** | 0.9 | 0.6 | 0.5 | 12.0 | 13.1 | 21.8 | 13.0 | 0.5 | 0.6 | 0.7 | 1.0 | 0.6 | 0.2 | 0.6 | 0.4 |
| Min-K%++ | 1.2 | 0.7 | 0.6 | 1.0 | 11.2 | 12.8 | 18.1 | 12.8 | 1.1 | **1.1** | **1.2** | **1.5** | 0.6 | 0.4 | 0.5 | 0.6 |
| DC-PDD | 0.9 | 0.4 | **1.2** | **1.4** | 10.8 | 11.3 | 9.8 | 10.7 | 0.4 | **1.1** | 0.6 | 1.1 | **1.5** | 0.8 | **1.3** | **1.3** |
| Ref | 0.9 | 0.8 | 0.7 | 0.9 | 13.4 | 10.9 | 17.9 | 4.7 | 0.6 | 0.5 | 0.7 | 1.2 | 0.8 | 0.9 | 0.2 | 0.5 |
| InfoRMIA1 | 0.8 | 1.0 | 0.7 | 1.0 | **14.7** | **17.5** | 21.0 | 14.9 | **1.2** | 0.9 | 0.7 | 0.7 | 1.0 | 0.7 | 0.5 | 0.7 |
| InfoRMIA2 | 1.1 | **1.3** | 0.9 | 1.0 | 13.0 | 13.8 | 18.7 | 14.5 | 0.4 | 0.5 | 0.5 | 0.6 | 1.2 | **1.2** | **1.3** | 0.8 |

| Method | ArXiv | | | | DM Mathematics | | | | HackerNews | | | | Average | | | |
|---|---|---|---|---|---|---|---|---|---|---|---|---|---|---|---|---|
| | 160M | 1.4B | 2.8B | 6.9B | 160M | 1.4B | 2.8B | 6.9B | 160M | 1.4B | 2.8B | 6.9B | 160M | 1.4B | 2.8B | 6.9B |
| Loss | 0.7 | 0.7 | 0.4 | 0.8 | 0.5 | 0.5 | 1.1 | 1.1 | 0.9 | 0.7 | 0.6 | 0.8 | 2.5 | 2.4 | 3.7 | 2.5 |
| Zlib | 0.5 | 0.2 | 0.4 | 0.7 | 1.1 | 0.9 | 0.9 | 0.6 | 0.6 | 1.0 | 1.0 | 1.0 | 2.7 | 3.0 | **4.1** | **2.9** |
| Min-K% | 0.3 | 0.3 | 0.4 | 0.7 | 0.8 | 0.6 | 0.2 | 0.4 | 0.7 | 0.9 | 0.7 | 1.1 | 2.3 | 2.4 | 3.6 | 2.4 |
| Min-K%++ | **1.1** | **1.9** | 1.2 | 1.4 | 1.0 | 1.0 | **1.2** | 1.0 | 0.7 | 0.5 | 1.1 | 0.7 | 2.4 | 2.6 | 3.4 | 2.7 |
| DC-PDD | 0.5 | 1.0 | 0.9 | 0.5 | 0.5 | 0.4 | 0.2 | 0.1 | 1.3 | 1.0 | 0.5 | 1.2 | 2.3 | 2.3 | 2.1 | 2.3 |
| Ref | 0.8 | 0.3 | 0.6 | 0.5 | **1.2** | 0.9 | 1.0 | 0.3 | 1.4 | 1.0 | 0.8 | 1.0 | 2.7 | 2.2 | 3.1 | 1.3 |
| InfoRMIA1 | 0.3 | 0.4 | 0.3 | 0.4 | 0.7 | **1.4** | 1.1 | **1.2** | 1.5 | **1.8** | 1.2 | 1.3 | **2.9** | **3.4** | 3.6 | **2.9** |
| InfoRMIA2 | 0.1 | 0.3 | 0.3 | 0.3 | **1.2** | 0.9 | **1.2** | 1.0 | **1.7** | 1.3 | 1.0 | **1.7** | 2.7 | 2.8 | 3.4 | 2.8 |

Table 7: TPR @0.1% FPR on MIMIR benchmark with deduped Pythia models when using the first step checkpoint of the target model as the reference.

| Method | Wikipedia | | | | Github | | | | Pile CC | | | | PubMed Central | | | |
|---|---|---|---|---|---|---|---|---|---|---|---|---|---|---|---|---|
| | 160M | 1.4B | 2.8B | 6.9B | 160M | 1.4B | 2.8B | 6.9B | 160M | 1.4B | 2.8B | 6.9B | 160M | 1.4B | 2.8B | 6.9B |
| Loss | 0.0 | **0.1** | **0.2** | 0.1 | 5.7 | 4.8 | 7.9 | 5.2 | 0.0 | **0.2** | 0.1 | 0.2 | **0.0** | 0.0 | 0.0 | 0.0 |
| Zlib | 0.1 | **0.1** | 0.1 | 0.1 | **8.4** | **5.7** | **8.6** | **6.9** | 0.0 | **0.2** | 0.2 | **0.3** | **0.0** | 0.0 | 0.0 | 0.0 |
| Min-K% | 0.0 | **0.1** | **0.2** | 0.1 | 5.7 | 4.8 | 8.0 | 4.9 | 0.0 | **0.2** | 0.1 | 0.2 | **0.0** | 0.0 | 0.0 | 0.0 |
| Min-K%++ | 0.0 | 0.0 | 0.1 | 0.0 | 6.1 | 3.5 | 4.6 | 2.1 | 0.1 | **0.2** | 0.1 | **0.3** | **0.0** | 0.0 | 0.0 | 0.0 |
| DC-PDD | 0.0 | 0.0 | 0.0 | 0.0 | 3.7 | 0.3 | 0.3 | 1.1 | 0.0 | 0.0 | **0.3** | 0.2 | **0.0** | 0.0 | 0.1 | 0.1 |
| Ref | 0.0 | **0.1** | 0.0 | 0.1 | 4.3 | 0.0 | 0.6 | 0.1 | **0.1** | 0.1 | 0.1 | 0.2 | **0.0** | 0.0 | 0.0 | 0.0 |
| InfoRMIA1 | **0.1** | **0.1** | **0.2** | **0.3** | 0.0 | 0.0 | 0.4 | 0.1 | **0.1** | 0.1 | 0.1 | 0.1 | **0.0** | **0.1** | **0.3** | **0.3** |
| InfoRMIA2 | 0.0 | 0.0 | 0.1 | 0.0 | 4.8 | 0.0 | 0.8 | 0.2 | **0.1** | 0.1 | 0.1 | 0.1 | **0.0** | 0.0 | 0.0 | 0.0 |

| Method | ArXiv | | | | DM Mathematics | | | | HackerNews | | | | Average | | | |
|---|---|---|---|---|---|---|---|---|---|---|---|---|---|---|---|---|
| | 160M | 1.4B | 2.8B | 6.9B | 160M | 1.4B | 2.8B | 6.9B | 160M | 1.4B | 2.8B | 6.9B | 160M | 1.4B | 2.8B | 6.9B |
| Loss | 0.1 | 0.0 | 0.0 | **0.1** | 0.0 | 0.0 | 0.2 | 0.0 | 0.1 | 0.0 | 0.0 | 0.0 | 0.8 | 0.7 | 1.2 | 0.8 |
| Zlib | 0.0 | 0.0 | 0.0 | 0.0 | 0.0 | 0.0 | 0.0 | 0.0 | 0.1 | **0.2** | **0.2** | 0.2 | 1.2 | 0.9 | 1.3 | 1.1 |
| Min-K% | 0.0 | 0.0 | 0.0 | 0.0 | 0.0 | 0.0 | 0.0 | 0.0 | **0.2** | 0.1 | 0.1 | 0.1 | 0.8 | 0.7 | 1.2 | 0.8 |
| Min-K%++ | 0.0 | 0.0 | 0.0 | 0.0 | 0.1 | 0.0 | **0.5** | 0.2 | **0.2** | 0.0 | 0.1 | 0.0 | 0.9 | 0.5 | 0.8 | 0.4 |
| DC-PDD | 0.0 | 0.0 | **0.2** | 0.0 | 0.0 | 0.0 | 0.0 | 0.0 | 0.0 | **0.2** | 0.1 | 0.0 | 0.5 | 0.1 | 0.1 | 0.2 |
| Ref | **0.2** | **0.1** | 0.0 | **0.1** | **0.2** | 0.0 | 0.3 | 0.0 | 0.1 | 0.0 | 0.0 | 0.3 | 0.7 | 0.0 | 0.1 | 0.1 |
| InfoRMIA1 | 0.1 | 0.0 | 0.0 | 0.0 | 0.1 | **0.1** | 0.3 | **0.5** | 0.0 | 0.0 | 0.0 | 0.0 | 0.1 | 0.1 | 0.2 | 0.2 |
| InfoRMIA2 | 0.0 | 0.0 | 0.0 | 0.0 | 0.0 | 0.0 | 0.0 | 0.0 | 0.1 | **0.2** | **0.2** | **0.4** | 0.7 | 0.0 | 0.2 | 0.1 |

Table 8: AUC results on MIMIR benchmark with deduped Pythia models when using the first step checkpoint of the target model as the reference.

| Method | Wikipedia | | | | Github | | | | Pile CC | | | | PubMed Central | | | |
|---|---|---|---|---|---|---|---|---|---|---|---|---|---|---|---|---|
| | 160M | 1.4B | 2.8B | 6.9B | 160M | 1.4B | 2.8B | 6.9B | 160M | 1.4B | 2.8B | 6.9B | 160M | 1.4B | 2.8B | 6.9B |
| Loss | 50.2 | 51.0 | 51.7 | 51.6 | 63.7 | 65.8 | 71.2 | 67.6 | 49.5 | 50.1 | 50.1 | 51.3 | 49.9 | 49.8 | 49.9 | 50.5 |
| Zlib | **51.0** | 51.8 | 52.4 | 52.3 | **65.6** | **67.2** | **72.2** | 68.8 | 49.6 | 50.2 | 50.3 | 51.2 | 50.0 | 50.0 | 50.0 | 50.6 |
| Min-K% | 48.6 | 50.6 | 51.6 | 51.4 | 63.6 | 65.9 | 71.4 | 68.0 | 50.0 | 51.0 | 50.5 | 51.9 | 50.4 | 50.2 | 50.4 | 51.0 |
| Min-K%++ | 47.7 | **52.3** | **53.7** | **52.4** | 61.4 | 65.7 | 70.7 | **69.1** | 49.8 | 51.1 | 49.9 | 51.7 | 50.9 | 50.6 | **51.2** | **52.3** |
| DC-PDD | 49.0 | 50.6 | 52.4 | 51.8 | 64.9 | 66.2 | 71.4 | 69.0 | 49.6 | **51.1** | **51.2** | **51.9** | 50.5 | **51.0** | 50.6 | 51.1 |
| Ref | 50.0 | 50.9 | 51.6 | 52.1 | 63.9 | 65.4 | 71.4 | 66.9 | 49.4 | 50.0 | 50.2 | 51.3 | 49.8 | 50.0 | 49.5 | 50.5 |
| InfoRMIA1 | 50.9 | 50.8 | 51.1 | 51.5 | 65.0 | 66.0 | 70.9 | 66.9 | 49.4 | 49.5 | 49.8 | 50.5 | 50.2 | 49.8 | 49.4 | 49.8 |
| InfoRMIA2 | 50.0 | 50.4 | 51.7 | 51.6 | 63.5 | 65.3 | 70.5 | 66.9 | **50.6** | 50.9 | 51.0 | 51.4 | **51.4** | 50.3 | 50.1 | 50.2 |

| Method | ArXiv | | | | DM Mathematics | | | | HackerNews | | | | Average | | | |
|---|---|---|---|---|---|---|---|---|---|---|---|---|---|---|---|---|
| | 160M | 1.4B | 2.8B | 6.9B | 160M | 1.4B | 2.8B | 6.9B | 160M | 1.4B | 2.8B | 6.9B | 160M | 1.4B | 2.8B | 6.9B |
| Loss | 50.7 | 51.4 | 51.9 | 52.5 | 49.0 | 48.6 | 48.3 | 48.4 | 49.2 | 50.4 | 51.2 | 51.7 | 51.8 | 52.4 | 53.5 | 53.4 |
| Zlib | 50.0 | 50.8 | 51.3 | 51.8 | 48.2 | 48.1 | 48.0 | 48.1 | 49.6 | 50.2 | 50.9 | 51.0 | 52.0 | 52.6 | 53.6 | 53.4 |
| Min-K% | 50.0 | 51.2 | 52.2 | 52.7 | 49.4 | 49.3 | 49.1 | 49.3 | 50.2 | 51.3 | 52.4 | 53.0 | 51.7 | 52.8 | 53.9 | 53.9 |
| Min-K%++ | 48.7 | 51.2 | **53.1** | 52.8 | **49.9** | **50.0** | 50.3 | 50.2 | **50.9** | 51.1 | 52.3 | 53.7 | 51.3 | **53.1** | 54.4 | **54.6** |
| DC-PDD | 50.4 | **52.0** | 52.9 | **52.9** | 49.0 | 49.3 | 49.8 | 49.7 | 50.7 | 51.8 | **53.0** | 53.9 | 52.0 | 53.1 | **54.5** | 54.3 |
| Ref | 50.3 | 51.3 | 51.8 | 52.3 | 48.8 | 49.0 | 48.7 | 48.2 | 49.1 | 50.5 | 51.4 | 51.5 | 51.6 | 52.5 | 53.5 | 53.3 |
| InfoRMIA1 | 50.3 | 51.2 | 51.2 | 51.6 | 48.0 | 47.7 | 47.8 | 47.8 | 50.4 | 50.7 | 51.1 | 51.2 | 52.0 | 52.3 | 53.0 | 52.8 |
| InfoRMIA2 | **50.8** | 51.0 | 51.6 | 52.2 | 49.0 | 49.1 | 49.0 | 48.8 | 50.4 | **52.0** | 52.3 | 52.3 | **52.2** | 52.7 | 53.7 | 53.3 |

## G.3 USING LATER CHECKPOINTS AS THE REFERENCE MODEL

In this section, we provide experiment results on MIMIR when we use a later checkpoint of the `Pythia-160m`. Given that the pretraining takes 143000 steps, we experimented with steps 10000 and 100000. Tables 9 to 14 are the respective results. Compared to Tables 3 to 5, the numbers show that using later checkpoints yields weaker attack performance. This aligns with our argument that we need the reference model to be as OUT as possible. Therefore, the earlier the checkpoint, the fewer training sequences it has seen, the better the attack performance is.

## G.4 MIMIR RESULTS WITH THE GPT-NEO FAMILY

The MIMIR benchmark also supports testing with models from the GPT-Neo family. However, since no intermediate checkpoints of GPT-Neos are released, we instead use the smallest pretrained model in the family, which is the `GPT-Neo-125m`, as the reference model, to attack the 1.3b and 2.7b models. Note that we have mentioned in the previous section that using a fully pretrained model as the reference model is bad, as the reference model is completely IN, instead of being OUT. Hence, the numbers in Tables 15 to 17 are just for curious readers.

## H TOKEN-BASED ANALYSIS

In this section, we provide some analytical results on the token-based interface on finetuned models on AG News and ai4privacy, when conducting offline token-level InfoRMIA with 4 reference models.

Table 9: AUC results on MIMIR benchmark for deduped Pythia models using the step 10k checkpoint of the 160m model. Note that the performance of InfoRMIA deteriorates as the reference model becomes less OUT when using a later checkpoint.

| Method | Wikipedia | | | Github | | | Pile CC | | | PubMed Central | | |
|---|---|---|---|---|---|---|---|---|---|---|---|---|
| | 160M | 1.4B | 2.8B | 160M | 1.4B | 2.8B | 160M | 1.4B | 2.8B | 160M | 1.4B | 2.8B |
| Loss | 50.2 | 51.0 | 51.7 | 63.7 | 65.8 | 71.2 | 49.5 | 50.1 | 50.1 | 49.9 | 49.8 | 49.9 |
| Zlib | **51.0** | 51.8 | 52.4 | **65.6** | **67.2** | **72.2** | 49.6 | 50.2 | 50.3 | 50.0 | 50.0 | 50.0 |
| Min-K% | 48.6 | 50.6 | 51.6 | 63.6 | 65.9 | 71.4 | 50.0 | 51.0 | 50.5 | 50.4 | 50.2 | 50.4 |
| Min-K%++ | 47.7 | **52.3** | **53.7** | 61.4 | 65.7 | 70.7 | 49.8 | 51.1 | 49.9 | **50.9** | 50.6 | **51.2** |
| DC-PDD | 49.0 | 50.6 | 52.4 | 64.9 | 66.2 | 71.4 | 49.6 | 51.1 | 51.2 | 50.5 | **51.0** | 50.6 |
| Ref | _49.9_ | _51.4_ | _53.0_ | 36.6 | 40.7 | 47.5 | _50.5_ | _52.1_ | _52.0_ | _50.3_ | _49.8_ | _50.1_ |
| Info-RMIA1 | 48.8 | 50.4 | 51.5 | _41.9_ | _43.3_ | _51.0_ | 48.5 | 51.1 | 51.0 | 49.7 | 49.0 | 49.0 |

| Method | ArXiv | | | DM Mathematics | | | HackerNews | | | Average | | |
|---|---|---|---|---|---|---|---|---|---|---|---|---|
| | 160M | 1.4B | 2.8B | 160M | 1.4B | 2.8B | 160M | 1.4B | 2.8B | 160M | 1.4B | 2.8B |
| Loss | **50.7** | 51.4 | 51.9 | 49.0 | 48.6 | 48.3 | 49.2 | 50.4 | 51.2 | 51.8 | 52.4 | 53.5 |
| Zlib | 50.0 | 50.8 | 51.3 | 48.2 | 48.1 | 48.0 | 49.6 | 50.2 | 50.9 | 52.0 | 52.6 | 53.6 |
| Min-K% | 50.0 | 51.2 | 52.2 | 49.4 | 49.3 | 49.1 | 50.2 | 51.3 | 52.4 | 51.7 | 52.8 | 53.9 |
| Min-K%++ | 48.7 | 51.2 | 53.1 | **49.9** | **50.0** | **50.3** | **50.9** | 51.1 | 52.3 | 51.3 | **53.1** | 54.4 |
| DC-PDD | 50.4 | 52.0 | 52.9 | 49.0 | 49.3 | 49.8 | 50.7 | 51.8 | 53.0 | **52.0** | 53.1 | **54.5** |
| Ref | 50.3 | _52.0_ | _53.1_ | _49.3_ | _48.6_ | _48.5_ | 50.4 | _52.9_ | _54.8_ | 48.2 | _49.6_ | _51.3_ |
| Info-RMIA1 | _50.5_ | 51.2 | 51.7 | 48.2 | 47.8 | 47.7 | _50.8_ | 52.5 | 53.4 | _48.3_ | 49.3 | 50.8 |

Table 10: TPR @0.1% FPR on MIMIR benchmark for deduped Pythia models using the step 10k checkpoint of the 160m model. Note that the performance of InfoRMIA deteriorates as the reference model becomes less OUT when using a later checkpoint.

| Method | Wikipedia | | | Github | | | Pile CC | | | PubMed Central | | |
|---|---|---|---|---|---|---|---|---|---|---|---|---|
| | 160M | 1.4B | 2.8B | 160M | 1.4B | 2.8B | 160M | 1.4B | 2.8B | 160M | 1.4B | 2.8B |
| Loss | 0.0 | 0.1 | 0.2 | 5.7 | 4.8 | 7.9 | 0.0 | 0.2 | 0.1 | 0.0 | **0.0** | 0.0 |
| Zlib | **0.1** | 0.1 | 0.1 | **8.4** | **5.7** | **8.6** | 0.0 | 0.2 | 0.2 | 0.0 | **0.0** | 0.0 |
| Min-K% | 0.0 | 0.1 | 0.2 | 5.7 | 4.8 | 8.0 | 0.0 | 0.2 | 0.1 | 0.0 | **0.0** | 0.0 |
| Min-K%++ | 0.0 | 0.0 | 0.1 | 6.1 | 3.5 | 4.6 | 0.1 | 0.2 | 0.1 | 0.0 | **0.0** | 0.0 |
| DC-PDD | 0.0 | 0.0 | 0.0 | 3.7 | 0.3 | 0.3 | 0.0 | 0.0 | 0.3 | 0.0 | **0.0** | 0.1 |
| Ref | 0.0 | **_0.2_** | **_0.3_** | _0.2_ | 0.0 | _0.8_ | **_0.2_** | _0.3_ | _0.4_ | _0.2_ | **_0.0_** | _0.0_ |
| Info-RMIA1 | **0.1** | 0.1 | 0.2 | 0.1 | _0.1_ | 0.5 | **_0.2_** | 0.2 | 0.3 | **_0.2_** | **_0.0_** | _0.0_ |

| Method | ArXiv | | | DM Mathematics | | | HackerNews | | | Average | | |
|---|---|---|---|---|---|---|---|---|---|---|---|---|
| | 160M | 1.4B | 2.8B | 160M | 1.4B | 2.8B | 160M | 1.4B | 2.8B | 160M | 1.4B | 2.8B |
| Loss | **0.1** | 0.0 | 0.0 | 0.0 | 0.0 | 0.2 | 0.1 | 0.0 | 0.0 | 0.8 | 0.7 | 1.2 |
| Zlib | 0.0 | 0.0 | 0.0 | 0.0 | 0.0 | 0.0 | 0.1 | **0.2** | **0.2** | **1.2** | **0.9** | **1.3** |
| Min-K% | 0.0 | 0.0 | 0.0 | 0.0 | 0.0 | 0.0 | **0.2** | 0.1 | 0.1 | 0.8 | 0.7 | 1.2 |
| Min-K%++ | 0.0 | 0.0 | 0.0 | **0.1** | 0.0 | **0.5** | **0.2** | 0.0 | 0.1 | 0.9 | 0.5 | 0.8 |
| DC-PDD | 0.0 | 0.0 | 0.2 | 0.0 | 0.0 | 0.0 | 0.0 | **0.2** | 0.1 | 0.5 | 0.1 | 0.1 |
| Ref | **_0.1_** | **_0.1_** | _0.0_ | _0.0_ | 0.0 | _0.0_ | _0.0_ | 0.0 | _0.1_ | _0.1_ | 0.1 | _0.2_ |
| Info-RMIA1 | **_0.1_** | **_0.1_** | _0.0_ | _0.0_ | **_0.1_** | _0.0_ | _0.0_ | _0.1_ | 0.0 | _0.1_ | _0.1_ | 0.1 |

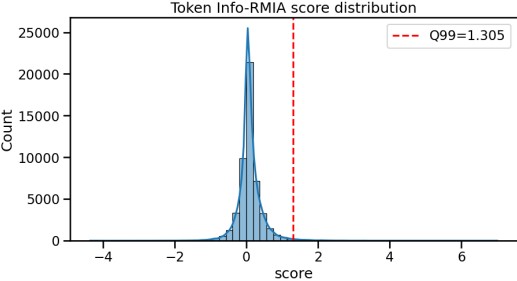

Figure 5: Distribution of token InfoRMIA scores on AG News dataset.

Table 11: TPR @1% FPR on MIMIR benchmark for deduped Pythia models using the step 10k checkpoint of the 160m model. Note that the performance of InfoRMIA deteriorates as the reference model becomes less OUT when using a later checkpoint.

| | Wikipedia | | | Github | | | Pile CC | | | PubMed Central | | |
|---|---|---|---|---|---|---|---|---|---|---|---|---|
| **Method** | 160M | 1.4B | 2.8B | 160M | 1.4B | 2.8B | 160M | 1.4B | 2.8B | 160M | 1.4B | 2.8B |
| Loss | 0.9 | 0.6 | 0.6 | 13.1 | 13.3 | 21.9 | 0.4 | 0.7 | 0.8 | 0.7 | 0.4 | 0.6 |
| Zlib | 1.3 | 0.7 | 0.8 | **14.3** | **16.9** | **24.0** | 0.7 | 0.7 | 0.9 | 0.3 | 0.4 | 0.5 |
| Min-K% | **1.4** | 0.9 | 0.6 | 12.0 | 13.1 | 21.8 | 0.5 | 0.6 | 0.7 | 0.6 | 0.2 | 0.6 |
| Min-K%++ | 1.2 | 0.7 | 0.6 | 11.2 | 12.8 | 18.1 | **1.1** | **1.1** | **1.2** | 0.6 | 0.4 | 0.5 |
| DC-PDD | 0.9 | 0.4 | **1.2** | 10.8 | 11.3 | 9.8 | 0.4 | **1.1** | 0.6 | 1.5 | 0.8 | 1.3 |
| Ref | 1.0 | 0.9 | 1.1 | 1.1 | 0.7 | 3.0 | 0.9 | **1.1** | **1.2** | **2.0** | **1.4** | **1.4** |
| Info-RMIA1 | 0.7 | **1.2** | 0.9 | 1.4 | 0.8 | 3.9 | 0.6 | 0.9 | 0.7 | 1.1 | **1.4** | 1.0 |

| | ArXiv | | | DM Mathematics | | | HackerNews | | | Average | | |
|---|---|---|---|---|---|---|---|---|---|---|---|---|
| **Method** | 160M | 1.4B | 2.8B | 160M | 1.4B | 2.8B | 160M | 1.4B | 2.8B | 160M | 1.4B | 2.8B |
| Loss | 0.7 | 0.7 | 0.4 | 0.5 | 0.5 | 1.1 | 0.9 | 0.7 | 0.6 | 2.5 | 2.4 | 3.7 |
| Zlib | 0.5 | 0.2 | 0.4 | **1.1** | 0.9 | 0.9 | 0.6 | 1.0 | 1.0 | **2.7** | **3.0** | **4.1** |
| Min-K% | 0.3 | 0.3 | 0.4 | 0.8 | 0.6 | 0.2 | 0.7 | 0.9 | 0.7 | 2.3 | 2.4 | 3.6 |
| Min-K%++ | 1.1 | **1.9** | 1.2 | 1.0 | 1.0 | **1.2** | 0.7 | 0.5 | 1.1 | 2.4 | 2.6 | 3.4 |
| DC-PDD | 0.5 | 1.0 | 0.9 | 0.5 | 0.4 | 0.2 | **1.3** | 1.0 | 0.5 | 2.3 | 2.3 | 2.1 |
| Ref | 1.2 | 1.5 | **1.6** | 0.6 | 1.0 | 1.1 | 0.5 | **1.5** | **1.6** | 1.0 | 1.2 | 1.6 |
| Info-RMIA1 | **1.9** | 1.2 | 1.2 | 0.8 | **1.6** | 1.1 | 1.1 | **1.5** | 1.5 | 1.1 | 1.2 | 1.5 |

Table 12: AUC results on MIMIR benchmark for deduped Pythia models using the step 100k checkpoint of the 160m model. Note that the performance of InfoRMIA deteriorates as the reference model becomes less OUT when using a later checkpoint.

| | Wikipedia | | | Github | | | Pile CC | | | PubMed Central | | |
|---|---|---|---|---|---|---|---|---|---|---|---|---|
| **Method** | 160M | 1.4B | 2.8B | 160M | 1.4B | 2.8B | 160M | 1.4B | 2.8B | 160M | 1.4B | 2.8B |
| Loss | 50.2 | 51.0 | 51.7 | 63.7 | 65.8 | 71.2 | 49.5 | 50.1 | 50.1 | 49.9 | 49.8 | 49.9 |
| Zlib | **51.0** | 51.8 | 52.4 | **65.6** | **67.2** | **72.2** | 49.6 | 50.2 | 50.3 | 50.0 | 50.0 | 50.0 |
| Min-K% | 48.6 | 50.6 | 51.6 | 63.6 | 65.9 | 71.4 | **50.0** | 51.0 | 50.5 | 50.4 | 50.2 | 50.4 |
| Min-K%++ | 47.7 | **52.3** | **53.7** | 61.4 | 65.7 | 70.7 | 49.8 | 51.1 | 49.9 | 50.9 | 50.6 | **51.2** |
| DC-PDD | 49.0 | 50.6 | 52.4 | 64.9 | 66.2 | 71.4 | 49.6 | 51.1 | 51.2 | 50.5 | **51.0** | 50.6 |
| Ref | 49.3 | 51.3 | 53.2 | 37.9 | 38.8 | 46.8 | 49.6 | **52.7** | **52.7** | **51.2** | 50.2 | 50.4 |
| Info-RMIA1 | 49.2 | 50.5 | 51.7 | 40.5 | 40.9 | 49.3 | 49.4 | 51.6 | 51.5 | 50.6 | 49.7 | 49.5 |

| | ArXiv | | | DM Mathematics | | | HackerNews | | | Average | | |
|---|---|---|---|---|---|---|---|---|---|---|---|---|
| **Method** | 160M | 1.4B | 2.8B | 160M | 1.4B | 2.8B | 160M | 1.4B | 2.8B | 160M | 1.4B | 2.8B |
| Loss | **50.7** | 51.4 | 51.9 | 49.0 | 48.6 | 48.3 | 49.2 | 50.4 | 51.2 | 51.8 | 52.4 | 53.5 |
| Zlib | 50.0 | 50.8 | 51.3 | 48.2 | 48.1 | 48.0 | 49.6 | 50.2 | 50.9 | 52.0 | 52.6 | 53.6 |
| Min-K% | 50.0 | 51.2 | 52.2 | 49.4 | 49.3 | 49.1 | 50.2 | 51.3 | 52.4 | 51.7 | 52.8 | 53.9 |
| Min-K%++ | 48.7 | 51.2 | **53.1** | 49.9 | **50.0** | **50.3** | 50.9 | 51.1 | 52.3 | 51.3 | **53.1** | 54.4 |
| DC-PDD | 50.4 | **52.0** | 52.9 | 49.0 | 49.3 | 49.8 | 50.7 | 51.8 | 53.0 | **52.0** | 53.1 | **54.5** |
| Ref | 48.2 | 51.9 | 53.0 | **50.2** | 48.9 | 48.8 | 50.2 | **52.9** | **55.3** | 48.1 | 49.5 | 51.5 |
| Info-RMIA1 | 49.3 | 51.0 | 51.7 | 49.2 | 48.0 | 47.8 | **51.6** | 52.5 | 53.6 | 48.5 | 49.2 | 50.7 |

Table 13: TPR @0.1% FPR on MIMIR benchmark for deduped Pythia models using the step 100k checkpoint of the 160m model. Note that the performance of InfoRMIA deteriorates as the reference model becomes less OUT when using a later checkpoint.

| Method | Wikipedia | | | Github | | | Pile CC | | | PubMed Central | | |
|---|---|---|---|---|---|---|---|---|---|---|---|---|
| | 160M | 1.4B | 2.8B | 160M | 1.4B | 2.8B | 160M | 1.4B | 2.8B | 160M | 1.4B | 2.8B |
| Loss | 0.0 | 0.1 | 0.2 | 5.7 | 4.8 | 7.9 | 0.0 | 0.2 | 0.1 | 0.0 | **0.0** | 0.0 |
| Zlib | **0.1** | 0.1 | 0.1 | **8.4** | **5.7** | **8.6** | 0.0 | 0.2 | 0.2 | 0.0 | **0.0** | 0.0 |
| Min-K% | 0.0 | 0.1 | 0.2 | 5.7 | 4.8 | 8.0 | 0.0 | 0.2 | 0.1 | 0.0 | **0.0** | 0.0 |
| Min-K%++ | 0.0 | 0.0 | 0.1 | 6.1 | 3.5 | 4.6 | 0.1 | 0.2 | 0.1 | 0.0 | **0.0** | 0.0 |
| DC-PDD | 0.0 | 0.0 | 0.0 | 3.7 | 0.3 | 0.3 | 0.0 | 0.0 | 0.3 | 0.0 | **0.0** | 0.1 |
| Ref | 0.1 | 0.2 | 0.3 | 0.1 | 0.0 | 0.5 | 0.2 | 0.4 | 0.5 | 0.0 | **0.0** | 0.0 |
| Info-RMIA1 | 0.1 | 0.2 | 0.1 | 0.0 | 0.0 | 0.2 | 0.1 | 0.3 | 0.3 | 0.1 | **0.0** | 0.0 |

| Method | ArXiv | | | DM Mathematics | | | HackerNews | | | Average | | |
|---|---|---|---|---|---|---|---|---|---|---|---|---|
| | 160M | 1.4B | 2.8B | 160M | 1.4B | 2.8B | 160M | 1.4B | 2.8B | 160M | 1.4B | 2.8B |
| Loss | 0.1 | 0.0 | 0.0 | 0.0 | **0.0** | 0.2 | 0.1 | 0.0 | 0.0 | 0.8 | 0.7 | 1.2 |
| Zlib | 0.0 | 0.0 | 0.0 | 0.0 | **0.0** | 0.0 | 0.1 | 0.2 | 0.2 | 1.2 | 0.9 | 1.3 |
| Min-K% | 0.0 | 0.0 | 0.0 | 0.0 | **0.0** | 0.0 | 0.2 | 0.1 | 0.1 | 0.8 | 0.7 | 1.2 |
| Min-K%++ | 0.0 | 0.0 | 0.0 | 0.1 | **0.0** | 0.5 | 0.2 | 0.0 | 0.1 | 0.9 | 0.5 | 0.8 |
| DC-PDD | 0.0 | 0.0 | 0.2 | 0.0 | **0.0** | 0.0 | 0.0 | 0.2 | 0.1 | 0.5 | 0.1 | 0.1 |
| Ref | 0.3 | 0.1 | 0.0 | 0.0 | **0.0** | 0.0 | 0.0 | 0.1 | 0.2 | 0.1 | 0.1 | 0.2 |
| Info-RMIA1 | 0.0 | 0.1 | 0.0 | 0.3 | **0.0** | 0.0 | 0.0 | 0.1 | 0.0 | 0.1 | 0.1 | 0.1 |

Table 14: TPR @1% FPR on MIMIR benchmark for deduped Pythia models using the step 100k checkpoint of the 160m model. Note that the performance of InfoRMIA deteriorates as the reference model becomes less OUT when using a later checkpoint.

| Method | Wikipedia | | | Github | | | Pile CC | | | PubMed Central | | |
|---|---|---|---|---|---|---|---|---|---|---|---|---|
| | 160M | 1.4B | 2.8B | 160M | 1.4B | 2.8B | 160M | 1.4B | 2.8B | 160M | 1.4B | 2.8B |
| Loss | 0.9 | 0.6 | 0.6 | 13.1 | 13.3 | 21.9 | 0.4 | 0.7 | 0.8 | 0.7 | 0.4 | 0.6 |
| Zlib | 1.3 | 0.7 | 0.8 | **14.3** | **16.9** | **24.0** | 0.7 | 0.7 | 0.9 | 0.3 | 0.4 | 0.5 |
| Min-K% | **1.4** | 0.9 | 0.6 | 12.0 | 13.1 | 21.8 | 0.5 | 0.6 | 0.7 | 0.6 | 0.2 | 0.6 |
| Min-K%++ | 1.2 | 0.7 | 0.6 | 11.2 | 12.8 | 18.1 | 1.1 | 1.1 | **1.2** | 0.6 | 0.4 | 0.5 |
| DC-PDD | 0.9 | 0.4 | 1.2 | 10.8 | 11.3 | 9.8 | 0.4 | 1.1 | 0.6 | **1.5** | 0.8 | **1.3** |
| Ref | 0.5 | 1.4 | 1.3 | 1.0 | 0.7 | 3.3 | 0.8 | 1.3 | 1.1 | 1.2 | 0.8 | 0.8 |
| Info-RMIA1 | 0.9 | 1.0 | 1.0 | 1.0 | 0.9 | 4.6 | 1.6 | 1.1 | 0.9 | 1.3 | **0.9** | 0.6 |

| Method | ArXiv | | | DM Mathematics | | | HackerNews | | | Average | | |
|---|---|---|---|---|---|---|---|---|---|---|---|---|
| | 160M | 1.4B | 2.8B | 160M | 1.4B | 2.8B | 160M | 1.4B | 2.8B | 160M | 1.4B | 2.8B |
| Loss | 0.7 | 0.7 | 0.4 | 0.5 | 0.5 | 1.1 | 0.9 | 0.7 | 0.6 | 2.5 | 2.4 | 3.7 |
| Zlib | 0.5 | 0.2 | 0.4 | 1.1 | 0.9 | 0.9 | 0.6 | **1.0** | 1.0 | **2.7** | **3.0** | **4.1** |
| Min-K% | 0.3 | 0.3 | 0.4 | 0.8 | 0.6 | 0.2 | 0.7 | 0.9 | 0.7 | 2.3 | 2.4 | 3.6 |
| Min-K%++ | 1.1 | **1.9** | **1.2** | 1.0 | **1.0** | **1.2** | 0.7 | 0.5 | 1.1 | 2.4 | 2.6 | 3.4 |
| DC-PDD | 0.5 | 1.0 | 0.9 | 0.5 | 0.4 | 0.2 | **1.3** | 1.0 | 0.5 | 2.3 | 2.3 | 2.1 |
| Ref | 0.8 | 1.3 | 1.2 | 1.0 | 1.0 | 0.6 | 0.4 | 0.9 | 1.1 | 0.8 | 1.1 | 1.3 |
| Info-RMIA1 | **1.3** | 1.1 | 1.0 | 1.3 | 1.0 | 0.7 | 0.9 | 1.0 | 2.0 | 1.2 | 1.0 | 1.5 |

Table 15: AUC results on MIMIR benchmark for GPT-Neo models. Note that the results here are only for curious readers, because the reference model is completely IN, breaking the offline attack assumption.

| Method | Wikipedia | | Github | | Pile CC | | PubMed Central | |
|---|---|---|---|---|---|---|---|---|
| | 1.3B | 2.7B | 1.3B | 2.7B | 1.3B | 2.7B | 1.3B | 2.7B |
| Loss | 51.0 | 51.3 | 68.1 | 69.9 | 50.0 | 50.4 | 49.6 | 49.8 |
| Zlib | 51.7 | 51.9 | 69.6 | 71.3 | 50.0 | 50.5 | 49.7 | 49.9 |
| Min-K% | 50.6 | 51.2 | 68.2 | 70.1 | 50.3 | 50.7 | 50.0 | 50.1 |
| Min-K%++ | 51.5 | **53.4** | 68.2 | 70.2 | 49.7 | 50.4 | 51.1 | **51.4** |
| DC-PDD | 50.7 | 51.2 | 69.8 | **71.5** | 50.6 | 50.7 | 50.7 | 50.3 |
| Ref | 51.3 | 51.7 | 46.9 | 48.0 | 52.5 | 52.9 | 49.3 | 50.0 |
| Info-RMIA1 | 50.9 | 50.8 | 48.4 | 49.8 | 51.5 | 51.8 | 48.6 | 49.1 |
| Info-RMIA2 | **52.1** | 53.2 | **69.9** | 71.3 | 52.4 | 52.0 | **51.2** | 51.2 |

| Method | ArXiv | | DM Mathematics | | HackerNews | | Average | |
|---|---|---|---|---|---|---|---|---|
| | 1.3B | 2.7B | 1.3B | 2.7B | 1.3B | 2.7B | 1.3B | 2.7B |
| Loss | 51.1 | 51.5 | 48.6 | 48.5 | 49.9 | 50.2 | 52.6 | 53.1 |
| Zlib | 50.6 | 51.0 | 48.1 | 48.1 | 50.1 | 50.2 | 52.8 | 53.3 |
| Min-K% | 51.2 | 51.7 | 49.2 | 49.2 | 51.4 | 51.7 | 53.0 | 53.5 |
| Min-K%++ | 52.1 | 52.1 | **49.6** | **49.7** | 50.8 | 51.5 | 53.3 | 54.1 |
| DC-PDD | 51.6 | 51.9 | 49.1 | 49.6 | 52.6 | 51.8 | 53.6 | 53.9 |
| Ref | 52.0 | **53.0** | 47.7 | 48.2 | **53.2** | **54.1** | 50.4 | 51.1 |
| Info-RMIA1 | 50.9 | 51.4 | 47.2 | 47.5 | 52.7 | 53.3 | 50.0 | 50.5 |
| Info-RMIA2 | **52.6** | 52.9 | 49.2 | 48.9 | 50.9 | 50.5 | **54.0** | **54.3** |

Table 16: TPR @0.1% FPR on MIMIR benchmark for GPT-Neo models. Note that the results here are only for curious readers, because the reference model is completely IN, breaking the offline attack assumption.

| Method | Wikipedia | | Github | | Pile CC | | PubMed Central | |
|---|---|---|---|---|---|---|---|---|
| | 1.3B | 2.7B | 1.3B | 2.7B | 1.3B | 2.7B | 1.3B | 2.7B |
| Loss | 0.1 | 0.1 | 4.7 | 6.0 | 0.1 | 0.2 | 0.0 | 0.0 |
| Zlib | 0.1 | 0.1 | **13.7** | **13.3** | **0.2** | 0.2 | 0.0 | 0.0 |
| Min-K% | 0.1 | 0.1 | 4.8 | 6.0 | **0.2** | 0.2 | 0.0 | 0.0 |
| Min-K%++ | 0.1 | 0.1 | 9.0 | 11.0 | **0.2** | 0.2 | 0.0 | 0.0 |
| DC-PDD | 0.0 | 0.0 | 0.1 | 0.1 | 0.1 | 0.2 | **0.2** | **0.1** |
| Ref | **0.2** | **0.3** | 0.1 | 0.4 | **0.2** | **0.4** | 0.0 | 0.0 |
| Info-RMIA1 | **0.2** | 0.2 | 0.2 | 0.3 | **0.2** | 0.3 | 0.0 | 0.0 |
| Info-RMIA2 | 0.0 | 0.0 | 0.4 | 1.1 | **0.2** | **0.4** | 0.0 | 0.0 |

| Method | ArXiv | | DM Mathematics | | HackerNews | | Average | |
|---|---|---|---|---|---|---|---|---|
| | 1.3B | 2.7B | 1.3B | 2.7B | 1.3B | 2.7B | 1.3B | 2.7B |
| Loss | **0.0** | **0.0** | 0.0 | 0.0 | 0.0 | 0.0 | 0.7 | 0.9 |
| Zlib | **0.0** | **0.0** | 0.0 | **0.1** | 0.0 | 0.0 | **2.0** | **2.0** |
| Min-K% | **0.0** | **0.0** | 0.0 | 0.0 | 0.0 | 0.0 | 0.7 | 0.9 |
| Min-K%++ | **0.0** | **0.0** | 0.5 | 0.0 | 0.2 | 0.0 | 1.4 | 1.6 |
| DC-PDD | **0.0** | **0.0** | 0.0 | 0.0 | 0.1 | **0.3** | 0.1 | 0.1 |
| Ref | **0.0** | **0.0** | 0.0 | **0.1** | 0.1 | 0.2 | 0.1 | 0.2 |
| Info-RMIA1 | **0.0** | **0.0** | 0.0 | **0.1** | 0.0 | 0.0 | 0.1 | 0.1 |
| Info-RMIA2 | **0.0** | **0.0** | 0.0 | **0.1** | **0.7** | 0.0 | 0.2 | 0.2 |

Table 17: TPR @1% FPR on MIMIR benchmark for GPT-Neo models. Note that the results here are only for curious readers, because the reference model is completely IN, breaking the offline attack assumption.

| | Wikipedia | | Github | | Pile CC | | PubMed Central | |
| --- | --- | --- | --- | --- | --- | --- | --- | --- |
| **Method** | 1.3B | 2.7B | 1.3B | 2.7B | 1.3B | 2.7B | 1.3B | 2.7B |
| Loss | 0.6 | 0.6 | 18.7 | 22.0 | 0.7 | 0.7 | 0.2 | 0.3 |
| Zlib | 0.5 | 0.6 | **20.1** | 22.2 | 0.7 | 0.8 | 0.3 | 0.3 |
| Min-K% | 0.6 | 0.5 | 18.9 | **22.3** | 0.7 | 0.5 | 0.3 | 0.6 |
| Min-K%++ | 0.6 | 0.5 | 19.0 | 21.9 | 0.8 | 0.9 | 0.5 | 0.5 |
| DC-PDD | 0.5 | 0.7 | 16.1 | 20.0 | 0.8 | 0.9 | **1.1** | 0.9 |
| Ref | 0.9 | **1.1** | 2.0 | 2.6 | **1.0** | 1.2 | 1.0 | **1.3** |
| Info-RMIA1 | 0.8 | 0.9 | 2.8 | 3.0 | **1.0** | 0.9 | 1.0 | 0.7 |
| Info-RMIA2 | **1.2** | 0.9 | 12.5 | 8.4 | 0.9 | **1.4** | 0.5 | 0.6 |

| | ArXiv | | DM Mathematics | | HackerNews | | Average | |
| --- | --- | --- | --- | --- | --- | --- | --- | --- |
| **Method** | 1.3B | 2.7B | 1.3B | 2.7B | 1.3B | 2.7B | 1.3B | 2.7B |
| Loss | 0.9 | 0.6 | 0.5 | 0.4 | 0.7 | 0.9 | 3.2 | 3.6 |
| Zlib | 0.4 | 0.5 | 0.5 | 0.5 | 0.5 | 0.5 | 3.3 | 3.6 |
| Min-K% | 0.7 | 0.5 | 0.2 | 0.2 | 0.7 | 0.8 | 3.2 | 3.6 |
| Min-K%++ | 1.2 | 1.4 | 1.1 | 1.1 | 0.8 | 0.8 | **3.4** | **3.9** |
| DC-PDD | 1.2 | 0.7 | 0.3 | 0.3 | 1.1 | **1.6** | 3.0 | 3.6 |
| Ref | **1.9** | **1.9** | 0.6 | 0.5 | 1.4 | 0.8 | 1.3 | 1.3 |
| Info-RMIA1 | 1.1 | 1.5 | 0.8 | 1.0 | **1.8** | **1.3** | 1.3 | 1.3 |
| Info-RMIA2 | 0.8 | 1.3 | **1.2** | **2.1** | 1.0 | 1.1 | **2.6** | **2.3** |

Table 18: Summary statistics of token membership scores grouped by entity type on AG News, sorted by mean scores. Top-1% scoring tokens are called "high" tokens. "n_high" is the number of high tokens, and high_rate" is the proportion of the tokens in each entity being high scoring. "None" entities are not nouns, which are unsurprisingly the majority.

| entity | count | mean_score | median_score | p95 | n_high | high_rate |
| --- | --- | --- | --- | --- | --- | --- |
| PERSON | 2225 | 0.156000 | 0.103894 | 0.847365 | 50 | 0.022472 |
| WORK_OF_ART | 107 | 0.135550 | 0.056464 | 0.695698 | 3 | 0.028037 |
| PRODUCT | 75 | 0.122673 | 0.068920 | 0.854701 | 0 | 0.000000 |
| FAC | 136 | 0.119761 | 0.067262 | 0.894820 | 1 | 0.007353 |
| LOC | 161 | 0.117694 | 0.078282 | 0.729102 | 1 | 0.006211 |
| TIME | 159 | 0.115637 | 0.080404 | 0.621918 | 1 | 0.006289 |
| ORG | 4624 | 0.113697 | 0.073791 | 0.729737 | 74 | 0.016003 |
| GPE | 1587 | 0.107486 | 0.064042 | 0.634189 | 20 | 0.012602 |
| QUANTITY | 79 | 0.107218 | 0.076787 | 0.625024 | 1 | 0.012658 |
| MONEY | 1139 | 0.103070 | 0.081028 | 0.464732 | 5 | 0.004390 |
| None | 36188 | 0.094550 | 0.056949 | 0.646802 | 327 | 0.009036 |
| EVENT | 188 | 0.094198 | 0.051614 | 0.684139 | 2 | 0.010638 |
| NORP | 391 | 0.086809 | 0.063955 | 0.619991 | 4 | 0.010230 |
| ORDINAL | 134 | 0.081780 | 0.039014 | 0.571259 | 0 | 0.000000 |
| CARDINAL | 720 | 0.072828 | 0.051281 | 0.558225 | 4 | 0.005556 |
| PERCENT | 123 | 0.052848 | 0.037016 | 0.407925 | 0 | 0.000000 |
| DATE | 1798 | 0.048768 | 0.031457 | 0.501247 | 6 | 0.003337 |
| LAW | 5 | 0.005115 | -0.096963 | 0.653592 | 0 | 0.000000 |
| LANGUAGE | 3 | -0.149360 | -0.130233 | 0.017094 | 0 | 0.000000 |

Table 19: Summary statistics of token membership scores grouped by their private/non-private status in the ai4privacy dataset.

| token | count | mean | std | min | 10% | 50% | 90% | max |
|---|---|---|---|---|---|---|---|---|
| Non-private | 147411.0 | 0.090224 | 0.303371 | -4.658639 | -0.088854 | 0.056127 | 0.304669 | 8.894643 |
| Private | 36340.0 | 0.076426 | 0.320213 | -4.478451 | -0.164783 | 0.048231 | 0.340272 | 7.925481 |

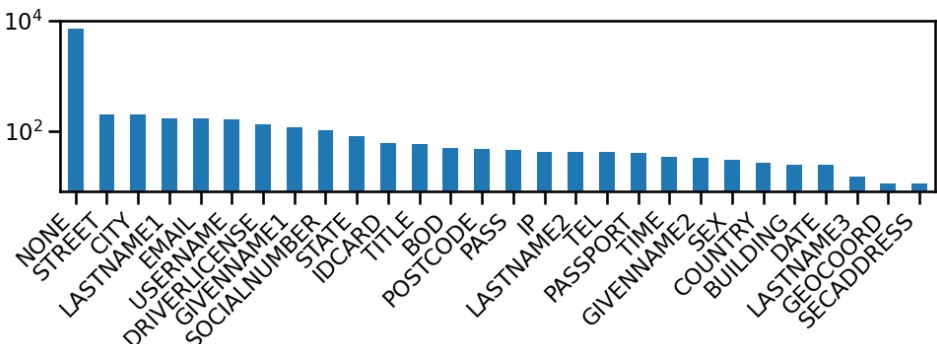

Figure 6: Distribution of high scoring tokens according to their types in ai4privacy dataset. The y-axis is the number of tokens in log scale.

**Seq 345** (sample_index=23060, avg=0.451, avg_priv=0.243)

1989. With the garimanjaly10@hotmail.com as her communication channel, she wields her P21WC0501915 like a badge of honor in this virtual world. Her 001 857 794-5305 is always at the ready for strategic discussions with fellow gamers. Armed with the opZ37^, she fearlessly navigates through quests and challenges, embodying strength and determination. Joining her on this gaming adventur

**Seq 358** (sample_index=14422, avg=0.440, avg_priv=0.182)

813", "entry_date": "2049-11-21T00:00:00", "entry_time": "6 AM", "location": "BS16 4EG", "behaviors": ["Practiced distress tolerance techniques", "Used interpersonal effectiveness skills", "Reviewed diary cards"], "reactions": ["Felt empowered by distress tolerance practice", "Successfully applied interpersonal skills in a difficult situation", "Identified patterns in diary card review"]} {"entry_id": 3, "user_id": "oflwnqgujwluzlvx09", "passport_id": "97

**Seq 234** (sample_index=8324, avg=0.367, avg_priv=0.228)

palasingam will present a case study highlighting the successful implementation of sustainable water management practices in the region. During the seminar, we will also delve into the challenges faced by water resource authorities, as illustrated by rodi.sprugasci's research on the impact of water scarcity on rural communities. gmmnodttjo66 will offer valuable insights into the legal implications of transbound

**Seq 113** (sample_index=17332, avg=0.353, avg_priv=0.180)

onet Comment: "Although uniforms restrict personal expression to some extent, they also create a sense of belonging and school pride that can positively impact the overall school atmosphere." 6. Username: Malou Comment: "I support school uniforms for their role in creating a level playing field for students from diverse socioeconomic backgrounds. It helps prevent discrimination based on clothing brands." 7. Username: Poulaillon Comment: "From a teacher's perspect

**Seq 233** (sample_index=15283, avg=0.334, avg_priv=0.132)

knowledge sharing sessions led by **Bogajo** to enhance curriculum coherence. **Professional Development Activities:** 1. **Pope** to lead a workshop on integrating technology into curriculum design. 2. Organize a conference on culturally responsive teaching strategies guided by **Lebada**. 3. Establish peer observation groups to promote best practices in **Denison** and **Clarkton** schools. 4. Implement a feedback mechanism utilizing **iyxmpwxbq

**Seq 491** (sample_index=81892, avg=0.331, avg_priv=nan)

Business Plan de e-commerce **Introduction** Le commerce électronique est en constante évolution, et pour réussir dans ce marché dynamique, il est essentiel d'avoir une stratégie solide et des objectifs clairs. Notre business plan pour notre entreprise e-commerce vise à définir nos actions et nos objectifs pour prospérer dans le secteur du commerce en ligne. **Stratégies clés** 1. **Segmentation du Marché**: Nous utiliserons les informations de nos clients pour diviser le marché en segments spécif

**Seq 434** (sample_index=20020, avg=0.330, avg_priv=nan)

Team Collaboration Platforms for Enhanced Pediatric Care Dear Team, In our continuous efforts to improve pediatric care services, we are excited to introduce a new team collaboration platform that will streamline our communication and enhance patient care outcomes. This platform aims to leverage technology to ensure efficient coordination among healthcare professionals and enhance the overall quality of care provided to our young patients. Key Features of

**Seq 38** (sample_index=130241, avg=0.326, avg_priv=0.129)

4. 23/03/1982 - Esperto legale sulle questioni dello spazio 5. 02/02/1965 - Esperta tecnica in comunicazioni satellitari 6. 18° febbraio 1972 - Rappresentante del settore degli investimenti spaziali 7. agosto/02 - Consulente di sicurezza spaziale DIRITTI E RESPONSABILITÀÀ Le parti concordano sulle seguenti clausole: - Le parti si impegnano a rispettare le normative spaziali nazionali e internazionali. - L'uso dello spettro satellitare sarà regola

**Seq 44** (sample_index=7758, avg=0.326, avg_priv=0.198)

ion Date": "21st November 2022", "Severance Pay": "$10,000", "Working Notice Period": "2 weeks", "Vacation Pay Owed": "$1,500", "Lump Sum Payment": "$5,000", "Return of Company Property Deadline": "7 days", "Confidentiality Clause": "Employee shall not disclose any confidential information after termination." } }}

**Seq 307** (sample_index=13138, avg=0.314, avg_priv=0.148)

CK] 2. **Communication with therapist:** Q6457998 - Female: [CLIENT FEEDBACK] 3. **Comfort level during the session:** Q6457998 - Female: [CLIENT FEEDBACK] 4. **Impact of the therapy on your well-being:** Q6457998 - Female: [CLIENT FEEDBACK] 5. **Suggestions for improvement:** Q6457998 - Female: [CLIENT FEEDBACK] --- *(Please repeat the above format for all remaining clients)*

Figure 7: Top-10 memorized sequences by a finetuned GPT-2 model on ai4privacy dataset, identified by InfoRMIA, ranked by sequence-based membership scores. Some of them do not even have any private tokens. Others have disproportionately small private token average scores.

**Top 10 sequences by average private-token score**

**Seq 76** (sample_index=131097, avg=0.207, avg_priv=0.667)

is": "Autismo moderato", "Specific_Behaviors": "Aggressione fisica, difficoltà nell'espressione verbale" } } }, { "SKARL.507225.SM.496": { "Skarlatos": { "Murteza": { "Diagnosis": "Disturbi dello spettro autistico non specificati", "Specific_Behaviors": "Rumore eccessivo, scarsa interazione sociale" } } } }, {

**Seq 377** (sample_index=1578, avg=0.212, avg_priv=0.537)

reness: Utilize resources and programs to educate residents on the risks of alcohol abuse.</li> <li>Support Services: Ensure access to counseling and support for individuals struggling with alcohol dependency.</li> <li>Regulation: Implement policies to control alcohol advertising and availability in the community.</li> </ul> <h2>Individual Responsibilities</h2> <p>Each member of the community, including [[Loélie]], [[Hatice]] and [[Jamrat]]

**Seq 216** (sample_index=99385, avg=0.172, avg_priv=0.510)

Betreff: Anfrage zur Rückmeldung - Appellationspraxis Sehr geehrte Damen und Herren, ich hoffe, diese Nachricht erreicht Sie im besten Wohlbefinden. Gerne würde ich Ihr wertvolles Feedback zu meiner juristischen Angelegenheit erhalten. Meine Kontaktdaten und weitere Informationen finden Sie unten. Datum der Anfrage: Mai 11., 2065 **Details des Antragstellers:** 1. Antragsteller: Mohammedberhan Geschlecht: A

**Seq 223** (sample_index=127813, avg=0.154, avg_priv=0.479)

nta e contribuisce a plasmare il futuro della nostra comunità. Le chiedo cortesemente di portare con sé il suo 465345506 come documento d'identità valido per partecipare alle elezioni. La sua presenza alle urne èè fondamentale per garantire una rappresentanza democratica e inclusiva. Resto a disposizione per qualsiasi domanda o chiarimento. Grazie per il suo coinvolgimento e impegno civico. Cordialmente, [Il tuo Nome] [Il tuo Ruolo] --------------------------------------

**Seq 257** (sample_index=169404, avg=0.176, avg_priv=0.463)

Management", "CompletionDate": "2023-01-25", "CertificateID": "170-Ruta de la Costa Vasca-Berriatua" }, { "EmployeeName": "ALICE", "EmployeeID": "24887", "TrainingProgram": "Strategic Planning in Healthcare", "CompletionDate": "2023-03-05", "CertificateID": "418-Calle Valle-Valderrueda Renedo de Valdetuéjar" } ] }

**Seq 583** (sample_index=95225, avg=0.099, avg_priv=0.435)

genden finden Sie die Protokolle sowie die detaillierten Aufzeichnungen der heutigen Vorlesung. 1. Teilnehmer STUDENT_J: - Benutzername: 1991M32 - Perspektive: Zweite Person Plural 2. Teilnehmer STUDENT_H: - Benutzername: vilares - Perspektive: Formale dritte Person Plural 3. Teilnehmer STUDENT_C: - Benutzername: 94kerrin - Perspektive: Formale zweite Person Plural 4.

**Seq 171** (sample_index=144775, avg=0.167, avg_priv=0.421)

"question_content": "In che modo gestisce i diritti di licenza per i contenuti digitali?" }, { "question_number": 2, "question_content": "Quali sono le tendenze emergenti che impattano la gestione dei diritti digitali?" } ] }, "applicant_7": { "title": "Infante", "designation": "Avvocato Senior", "country": "Switzerland", "specialty": "Online Privacy Law", "questions": [ {

**Seq 447** (sample_index=78987, avg=0.152, avg_priv=0.407)

Commentaires du suivi du projet - Service d'oncologie pédiatrique, Hôpital de Genève Chers collègues, Nous souhaitons partager avec vous les derniers commentaires concernant le suivi du projet en oncologie pédiatrique. Veuillez trouver ci-dessous un résumé détaillé des contributions de chaque membre de l'équipe : 1. Nom : Kushch Adresse IP : 230.233.131.185 Commentaire : Nous avons besoin de plus de données pour évaluer la progression du tra

**Seq 480** (sample_index=59655, avg=0.113, avg_priv=0.402)

```yaml health_promotion_campaigns: campaign1: id: "Campaña de Promoción de la Salud para Todos" id: "Involucra a la comunidad en hábitos saludables y prevención de enfermedades." id: "Todos los miembros de la comunidad" id: "Fomentar la adopción de un estilo de vida saludable y conciencia sobre salud preventiva" id: "Talleres educativos, sesiones informativas, actividades al aire libre" id: "15-07-2023" id: "15-08-2023" id: "9:55" id: "España" ```

**Seq 417** (sample_index=165879, avg=0.201, avg_priv=0.400)

Buen día, me gustaría participar en el estudio de investigación que han mencionado en clase. Quedo atento a cualquier requerimiento adicional. --- **Correo 2** Asunto: Horario de Clases Remitente: Masculino N° Documento: V27969571 Mensaje: Hola, desearía conocer si es posible modificar mi horario de clases, ya que se me han presentado conflictos con una actividad extracurricular. Quedo a la espera de su pronta respuesta. --- **Cor

Figure 8: Top-10 sequences that have the highest average scores of private tokens by a finetuned GPT-2 model on ai4privacy dataset, identified by InfoRMIA. Note that their sequence averages are much smaller compared to those in Figure 7, and the average private token scores. This is aligned with our intuition that signals from private tokens can get diluted in long texts.

