# OpenReview forum: "Information-Theoretic Membership Inference for Granular Quantification of Memorization"
_ICLR.cc/2026/Conference — ICLR 2026 Poster_

### Official Review · Reviewer_pY12 · 2025-10-25

**Soundness:** 2
**Presentation:** 2
**Contribution:** 3
**Rating:** 4
**Confidence:** 3

**Summary:**

This paper studies data memorization issues of LLMs through membership inference attacks. The paper proposes InfoRMIA, a membership inference attack stronger than the Robust Membership Inference Attack (RMIA). InfoRMIA extends membership inference attacks from the sentence level to the token level, providing a more fine-grained analysis of privacy.  Experimental results suggest that their method achieves new state-of-the-art performance and offers a more detailed perspective on memory leakage in large language models.

**Strengths:**

1.	The paper proposes studying privacy from a token-level perspective, which offers finer granularity compared to the traditional sequence-level analysis.

2. Experimental results show that  InfoRMIA achieves a new state-of-the-art performance, surpassing prior baseline RMIA.

**Weaknesses:**

1.	The theoretical derivations in the paper are difficult to follow and would benefit from clearer explanations and additional intuition. For example, in Section 4.2, it is hard to understand what the authors are trying to emphasize, as the section directly presents a formula without sufficient explanation or intuition.

2.	The paper lacks a sufficient explanation of the experimental setup. While Section 5 introduces the analysis tools used, Section 6 directly presents the results and conclusions without clarifying the chosen baselines or model configurations.

3.	It would be better to include more experimental results to substantiate the claimed SOTA performance. Although Tables 1 and 2 show that InfoRMIA generally outperforms RMIA, the advantages appear marginal in several cases (e.g., AUC 0.822 vs. 0.821 on the ai4privacy dataset). Moreover, some results in the appendix suggest that InfoRMIA’s performance is not entirely stable. Providing additional experiments and analyses would help strengthen the paper’s main claim.

**Questions:**

1. I may miss some paper details. Could the authors clarify the experimental setup in more detail, including dataset splits, model architectures, and training configurations? How are the auditor tools used?

2. Could the authors include additional experiments to better demonstrate the robustness and consistency of InfoRMIA’s performance?

3. Could the authors discuss cases where InfoRMIA underperforms or performs similarly to RMIA? Will LiRA be considered as one extra baseline?

4. Could the authors provide more intuition behind the theoretical derivations, especially in Section 4.2?

---

> ### Author Response · Authors · 2025-11-24
> **Rebuttal to Reviewer pY12 [1/2]**
>
> Thank you very much for your review. Here are our responses.
> - > [Q1][W2] "... clarify the experimental setup..."
>
> Thank you for the suggestion. The setup is briefly explained in Appendix C, but we acknowledge that it may still be confusing. We will add more details in the main text and insert clearer references to the corresponding appendix sections.
>
> In short, the experiments conducted with Privacy Meter use the tool’s default configurations. We have provided the exact link in the appendix. Privacy Meter trains target and reference models on a randomly selected half of the specified dataset. The model architectures are included in the default configurations: GPT-2 for AG News, WideResNet-28-2 for CIFAR-10, and an MLP for Purchase100. We will include the full commands used to run these experiments in the updated appendix.
>
> For Table 2, we will elaborate on the setup as well. Similarly, we fine-tune the target and reference models on a randomly chosen half of each dataset using GPT-2 as the backbone. We will release the full code upon acceptance, which should help readers better understand the setup.
>
> - > [Q2][W3] "... better demonstrate the robustness and consistency of InfoRMIA...", results in the appendix suggest that InfoRMIA’s performance is not entirely stable"
>
> We will add LiRA to both Table 1 and Table 2. The RMIA paper already provides a comprehensive evaluation demonstrating RMIA’s superiority over all other baselines, including LiRA. Therefore, we originally focused on comparisons against RMIA. However, we agree that including LiRA here would make the paper more self-contained and informative. We will upload an updated version of the manuscript soon. For reference, we attach the updated Table 2 for your preview. LiRA appears to be inferior to both RMIA and InfoRMIA, which supports our SOTA claim.
>
> Regarding the results in the appendix: these correspond to pretrained LLMs, which are widely known to be very challenging for MIAs. Across recent methods such as MinK++, ReCaLL, and DC-PDD, no method consistently performs best across all domains and model sizes. This is largely due to uneven memorization across datasets and differences in model architectures. This aligns with a key property of pretrained LLMs: they are trained for only one epoch and therefore do not memorize their entire training corpus. Because membership inference fundamentally relies on memorization, all attacks naturally perform poorly under this setup. Consequently, when discussing SOTA MIAs, the community typically focuses on settings where the model has converged. Hence, we put it in the appendix.
>
> We also want to emphasize that this is the less desirable setting for InfoRMIA due to the lack of quality reference models for pretrained LLMs, as mentioned in the paper, and we used the token-level version for practicality concerns. Yet, we show that (token-level) InfoRMIA is one of the best methods on average.
>
> - > [W3] "Providing additional experiments and analyses "
>
> We have evaluated InfoRMIA on nearly all standard benchmarks: tabular, image, text datasets, as well as both finetuned and pretrained LLMs. We believe the current experiments provide strong evidence that InfoRMIA achieves new SOTA performance for traditional neural networks and finetuned LLMs, surpassing RMIA.
>
> For pretrained LLMs, there are limited publicly accepted benchmarks. Based on the MIMIR results, InfoRMIA is one of the best methods. Given that token-level InfoRMIA also provides fine-grained memorization analysis for LLMs, we believe the paper offers significant value to the community (especially after adding LiRA). Still, we will expand our discussion and analyses based on all reviewer feedback.
>
> - > [Q3] "Will LiRA be considered as one extra baseline?"
>
> Thank you for the suggestion. We will update Tables 1 and 2 to include LiRA. We attach the updated Table 2 for convenience. Consistent with the RMIA paper, we observe that LiRA is dominated by both RMIA and InfoRMIA.
>
> | Dataset        | FT Epochs | RMIA AUC  | RMIA TRP@FPR | InfoRMIA AUC | InfoRMIA TRP@FPR | InfoRMIA (token) AUC | InfoRMIA (token) TRP@FPR | LiRA AUC | LiRA TRP@FPR |
> | -------------- | --------- | --------- | ------------ | ------------ | ---------------- | -------------------- | ------------------------ | -------- | ------------ |
> | **AG News**    | 1         | 0.839     | 0.00%        | **0.843**    | **23.0%**        | 0.836                | 20.2%                    | 0.795    | 3.6%         |
> |                | 4         | **0.945** | 0.00%        | **0.945**    | 16.2%            | 0.942                | **20.6%**                | 0.882    | 9.00%        |
> | **ai4privacy** | 1         | 0.643     | 6.6%         | **0.644**    | **10.6%**        | 0.620                | 9.0%                     | 0.630    | 3.8%        |
> |                | 4         | 0.821     | 26.0%        | **0.822**    | **27.2%**        | 0.804                | 23.2%                    | 0.782    | 10.4%         |

---

> > ### Author Response · Authors · 2025-11-24
> > **Rebuttal to Reviewer pY12 [2/2]**
> >
> > - > [Q2][Q3] "the advantages appear marginal in several cases", ""Could the authors discuss cases where InfoRMIA underperforms or performs similarly to RMIA?
> >
> > Interesting question, but infoRMIA would never underperform compared to RMIA, theoretically and empirically. We have explained the reason in Sec 3.3 why InfoRMIA is mathematically better than RMIA, but we would elaborate more on the factors affecting the gap. The key factor determining the gap between RMIA and InfoRMIA is the coarseness of RMIA’s population-based signals. RMIA uses percentile-based binning over the signals computed from Z. When the distribution of population signals is finely spread, less precision is lost through binning, and RMIA approaches the performance of InfoRMIA. In contrast, InfoRMIA uses a continuous statistic and does not suffer from these discretization effects.
> >
> > This also explains why RMIA requires a large population set for stable performance. Thus, cases where RMIA appears close to InfoRMIA simply reflect better-behaved population distributions—not that InfoRMIA underperforms.
> >
> > - > [Q4] "Could the authors provide more intuition behind the theoretical derivations, especially in Section 4.2?"
> >
> > Sure. We think your confusion is due to our omission of a problem statement, which is the standard one for membership inference. We will add it in the main text. The equations themselves are mostly math manipulations, so we think you want more explanations on what we are computing. So in membership inference, the most standard and optimal way to formulate the problem is via hypothesis testing (Eqn 4), on which we conduct a likelihood ratio test. The test statistic is often known as the membership score, is the core of each attach technique. Hence, in Eqn 3, we propose InfoRMIA's test statistic. We explained the intuition of its formulation in the paper, and in Eqn 3, we derived its equivalent forms. In Section 4.2, we took the same test statistic, but the $z$ is other token(s) instead of population data, we explained that $\sum_z p(z)=1$. This removes the need of a normalization step needed in Eqn 3 to get to the KL divergence form if we include $x$ itself in $Z$. Hence, we show from Eqn 6 to 9 that the inclusion of $x$ to $Z$ could lead to the KL divergence form without additional normalization. All in all, these are all equalities, which are different ways one can compute the test statistic. We will add "Test Statistic" to Eqn 6 to emphasize what we are computing.
> >
> > We hope we have address all your concerns. Please let us know if you have more questions.

---

> > > ### Comment · Reviewer_pY12 · 2025-11-26
> > >
> > > Dear Authors,
> > >
> > > Thanks for your clarifications about the experimental setups. I have raised my review score accordingly.
> > >
> > > Still, even though I am familiar with MIA, I find it hard to read the paper and understand the evaluation settings. I would like to increase my score to 8 if the details can be given in the main paper, and full details are available in the Appendix.
> > >
> > > Wish you good luck!

---

> > > > ### Author Response · Authors · 2025-11-27
> > > >
> > > > Dear Review pY12
> > > >
> > > > Thank you for pointing this out. We agree that including the necessary setup and evaluation details would improve the reading experience of our paper. We have updated the paper with a dedicated section (the new Section 6.1) explaining the setup, evaluation pipeline, and model information. We also added a brief overview at the beginning of Section 6 for better organization.
> > > >
> > > > If you think there are still important pieces of information missing, please let us know. We are more than happy to further improve the paper’s quality.
> > > >
> > > > 24791 authors

---

### Official Review · Reviewer_zEAx · 2025-10-30

**Soundness:** 3
**Presentation:** 3
**Contribution:** 3
**Rating:** 6
**Confidence:** 3

**Summary:**

This paper introduces InfoRMIA, a novel membership inference attack (MIA) method for large language models (LLMs) that significantly improves upon the original Range Membership Inference Attack (RMIA). The authors propose a theoretically grounded approach based on information theory principles, using a continuous test statistic that eliminates the need for the hyperparameter γ required in RMIA. The paper demonstrates that InfoRMIA provides higher precision in membership scores and reduces computational overhead. Additionally, the authors conduct token-level analysis, revealing that personally identifiable information (PII) such as names of people and artworks is disproportionately more memorized by LLMs compared to other token types. The paper shows that sequence-level privacy metrics like AUC may overestimate privacy risks, as they often reflect memorization of non-private content, making them poor indicators of true privacy risk.

**Strengths:**

S1: The paper provides a strong theoretical foundation for InfoRMIA, deriving a continuous test statistic based on information theory principles that is more precise than the discrete statistic used in RMIA.

S2: The token-level analysis is a significant contribution that reveals insights not captured by aggregate metrics like AUC, showing that PII (personally identifiable information) is disproportionately more memorized by LLMs.

S3: The paper effectively demonstrates that sequence-based frameworks overestimate privacy risks by showing that the most memorized sequences often contain no private tokens (Figure 2).

S4: The experimental evaluation is comprehensive, showing InfoRMIA's superiority over RMIA across multiple standard datasets (Purchase100, CIFAR-10, AG News) and models.

S5: The paper clearly explains why AUC alone is a poor indicator of true privacy risk in LLMs, providing a valuable insight for the privacy research community.

**Weaknesses:**

W1: The paper lacks sufficient analysis of InfoRMIA's performance across different LLM architectures and sizes. The experiments are limited to a few standard datasets and models, which may not fully represent the diversity of real-world LLMs.

W2: While the paper claims "reduced computational complexity," it does not provide specific quantitative comparisons of computational costs between InfoRMIA and RMIA, making it difficult to assess the practical significance of this claim.

W3: The token-level analysis is limited to only two datasets (AG News and ai4privacy), which may not be sufficient to generalize the findings about PII memorization patterns across different domains and languages.

W4: The paper does not discuss potential defense mechanisms against InfoRMIA or how the insights from the token-level analysis could inform the development of more privacy-preserving training methods.

W5: The counterintuitive finding that private tokens sometimes show lower membership scores than non-private tokens (in ai4privacy) is not sufficiently explored, leaving an important observation underdeveloped.

**Questions:**

Q1: Could you provide more detailed computational complexity analysis comparing InfoRMIA and RMIA, including execution time and memory usage measurements across different dataset sizes? This would strengthen the claim about reduced computational overhead.

Q2: Would it be possible to expand the token-level analysis to include more diverse datasets across different languages and domains to better generalize the findings about PII memorization patterns?

Q3: Could you investigate the reason behind the observation that private tokens sometimes show lower membership scores than non-private tokens in ai4privacy? This could have important implications for privacy risk assessment.

Q4: How might the insights from your token-level analysis be used to develop more effective privacy-preserving training methods or defenses against membership inference attacks?

Q5: Could you compare InfoRMIA with other state-of-the-art membership inference attacks beyond RMIA to better position the contribution of your work in the broader field?

Q6: The paper mentions "token-level framework" but doesn't clearly explain the methodology for token-level analysis. Could you provide more details on how the token-level scores are calculated and how they relate to the sequence-level scores?

---

> ### Author Response · Authors · 2025-11-24
> **Rebuttal to Reviewer zEAx [1/2]**
>
> Thank you very much for your review. Here are our responses.
> - > [Q1][W2] "computational complexity analysis"
>
> Thank you for pointing this out. We will add consolidated pseudocode comparing RMIA, InfoRMIA, and token-level InfoRMIA. This makes the efficiency gain clearer: the improvement comes from removing, or greatly reducing, the population dataset $Z$, where population signals $p(z)$ have to be computed for all $z \in Z$. This is why InfoRMIA incurs lower computational cost.
>
> ### Algorithm 1: MIA Score Computation (RMIA / InfoRMIA / Token-Level InfoRMIA)
>
> Input:
>     - Target model θ
>     - Reference models Θ (size k)
>     - Query x
>     - Hyperparameters γ, a
>     - Population dataset Z (for RMIA and InfoRMIA)
>
> Output:
>     - MIA score Score_MIA(x; θ)
>
> ````python
> 1:  Compute p(x|θ) and p(x|θ_r) for all θ_r in Θ
>
> 2:  p_out(x) ← (1/k) * sum_{θ_r in Θ} p(x|θ_r)
>
> 3:  p(x) ← 0.5 * ((1 + a) * p_out(x) + (1 - a))
>
> 4:  Ratio_x ← p(x|θ) / p(x)
>
> 5:  if Token-Level InfoRMIA then
> 6:       Take all non-ground-truth tokens as z        (See Sec. 4.2)
> 7:  else
> 8:       Take population samples z from dataset Z
> 9:  end if
>
> 10: for each z do
> 11:      Compute p(z|θ) and p(z|θ_r) for all θ_r in Θ
> 12:      p(z) ← (1/k) * sum_{θ_r in Θ} Pr(z|θ_r)
> 13:      Ratio_z ← p(z|θ) / p(z)
> 14: end for
>
> 15: if RMIA then
> 16:      Score_MIA(x; θ) ← (1/|Z|) * sum_z I( Ratio_x / Ratio_z ≥ γ )
> 17: else
> 18:      Score_MIA(x; θ) ← log(Ratio_x)
>                           − sum_z p(z) * log(Ratio_z)
> 19: end if
> ````
>
> - > [Q2][W3] " to better generalize the findings about PII memorization patterns"
>
> Section 5 is intended as a demo of the token-level inspection tool rather than a claim of universal PII patterns. The two datasets illustrate two contrasting worlds: AG News where PIIs are more memorized and ai4privacy where private tokens have smaller average memorization. We think these these opposite behaviors are sufficient to demonstrate why sequence-level metrics (like AUC) can be misleading and why token-level inspection provides more faithful insight. We will clarify that the purpose is methodological demonstration, not generalization.
>
> - > [Q3][W5] "investigate the reason behind the observation that private tokens sometimes show lower membership scores"
>
> Private tokens in ai4privacy (e.g., passwords, IPs) are usually random, high-entropy strings that occur extremely rarely. Memorizing these long-tailed outliers (in the sense of Feldman) requires the model to fully converge, which typically takes hundreds of epochs for small CNNs/MLPs. This is practically impossible for LLMs. Our target model was finetuned for 4 epochs. Hence, the private tokens were not entirely learned by the LLM, thus the smaller membership scores. We will include this explanation in the main text.
>
> - > [Q4][W4] "develop more effective privacy-preserving training methods or defenses"
>
> Since membership inference fundamentally stems from memorization, differentially private training (DP-SGD) remains an effective universal defense, including against InfoRMIA. However, our token-level analysis enables much more precise identification of what to unlearn. Instead of unlearning entire sentences (which may contain important non-sensitive content), token-level scores can guide targeted machine unlearning focused only on subsequences containing sensitive tokens. We will expand our discussion accordingly.

---

> > ### Author Response · Authors · 2025-11-24
> > **Rebuttal to Reviewer zEAx [2/2]**
> >
> > - > [Q5] "compare InfoRMIA with other state-of-the-art membership inference attacks"
> >
> > Thank you for the suggestion. We originally omitted LiRA because the RMIA paper had already shown RMIA’s superiority over previous methods. However, we agree that adding LiRA improves self-containment. We will update Tables 1 and 2 with LiRA results. Below is the updated Table 2 (preview). As in RMIA, LiRA is dominated by RMIA and therefore also by InfoRMIA.
> >
> > | Dataset        | FT Epochs | RMIA AUC  | RMIA TRP@FPR | InfoRMIA AUC | InfoRMIA TRP@FPR | InfoRMIA (token) AUC | InfoRMIA (token) TRP@FPR | LiRA AUC | LiRA TRP@FPR |
> > | -------------- | --------- | --------- | ------------ | ------------ | ---------------- | -------------------- | ------------------------ | -------- | ------------ |
> > | **AG News**    | 1         | 0.839     | 0.00%        | **0.843**    | **23.0%**        | 0.836                | 20.2%                    | 0.795    | 3.6%         |
> > |                | 4         | **0.945** | 0.00%        | **0.945**    | 16.2%            | 0.942                | **20.6%**                | 0.882    | 9.00%        |
> > | **ai4privacy** | 1         | 0.643     | 6.6%         | **0.644**    | **10.6%**        | 0.620                | 9.0%                     | 0.630    | 3.8%        |
> > |                | 4         | 0.821     | 26.0%        | **0.822**    | **27.2%**        | 0.804                | 23.2%                    | 0.782    | 10.4%         |
> >
> > - >[Q6] provide more details on how the token-level scores are calculated and how they relate to the sequence-level scores
> >
> > Thank you for your advice. In the pseudocode, we will highlight that the $x$ for token-level InfoRMIA is a prefixal substring. The score, thus, will be the membership score at the specific token completion. We will also clarify that the sequence-level score is simply an aggregation (e.g., averaging or min-k) of all token-level membership scores across the sequence.
> >
> > - > [W1] "The experiments are limited to a few standard datasets and models, which may not fully represent the diversity of real-world LLMs"
> >
> > As you mentioned in S4, we had a comprehensive evaluation on diverse and multiple datasets. For LLMs, due to the availability of reliable benchmarks, we only have MIMIR (WikiMIA was a popular benchmark but researchers have shown it suffers significant distributional shift). To maintain fair comparison with existing MIA methods, we evaluate on the standard benchmarks used in the literature. That said, we will add additional results on the GPT-Neo family in the revision to broaden coverage.
> >
> > Hope we have address all of your concerns.

---

### Official Review · Reviewer_hMRp · 2025-11-01

**Soundness:** 3
**Presentation:** 3
**Contribution:** 3
**Rating:** 4
**Confidence:** 3

**Summary:**

This paper proposes a fine-grained member inference attack, specifically at the token level, called InfoRMIA, which calculates the leakage of individual tokens and aggregates them to the sequence level to achieve stronger inference capabilities.This paper conducts an in-depth analysis of the existing best MIA (RMIA), and then proposes a more principled statistical test method (InfoRMIA), and analyzes its advantages over RMIA This paper proposes a token-level MIA framework to better quantify the memory and information leakage of LLMS with finer granularity and more meaningful analysis

**Strengths:**

- Clear theoretical framework construction
- Clear and concise formula and algorithm expressions

**Weaknesses:**

- Lack of more effective aggregation methods
- No method for distinguishing between sensitive and non-sensitive tokens was proposed
- The truly sensitive information is diluted or concealed

**Questions:**

The article proposes a new attack method, InfoRMIA, and the author clearly explains the reasons why it is superior to RMIA, reducing the dependence on the size of the dataset Z and improving the accuracy by using Bayesian factors. The author pointed out the shortcomings of the traditional MIA, such as information loss caused by global averaging, and proposed token-level member signal analysis, which is of great practical significance and value. The author pointed out that under the traditional MIA test, tokens with high member scores are more likely to contain less sensitive information, thereby highlighting the limitations and flaws of the AUC metric, that is, AUC to a greater extent reflects the member signals of non-sensitive tokens. These issues are all very valuable and worthy of in-depth consideration. Moreover, the author expounds these issues with clear logic and evidence, which is highly persuasive

However, I do have the following comments:

- Lack of more effective aggregation methods
On page 2, the author claims that aggregating from token-level metrics to sequence-level metrics would make it less significant for privacy assessment. However, the method used when migrating from the token-level to the sequence-level in the author's proposed approach is also a common aggregation method. Is this somewhat contradictory
- Lack of a mechanism to distinguish tokens from the opponent's perspective
On page 5, the author states that the work of this paper can precisely locate information leakage, and the scenario proposed by the author is that users observe the member scores of each token and check whether these tokens are sensitive. I don't think this is the most appropriate approach, because in member inference attacks, the adversary needs to filter and confirm which tokens are truly sensitive, and this process should be completed automatically, as the amount of data could be extremely large. This requires the formulation of some kind of recognition algorithm to complete the screening of truly sensitive tokens. For tokens that will not disclose privacy, even if their member scores are high, they should still be excluded
- The truly sensitive information is diluted or concealed(The same as the first point)
On page 6, the author mentioned better aggregation methods, but glossed over them due to their limitations and used common aggregation methods such as averaging and min-k. However, before screening out the truly sensitive tokens, simply averaging the tokens in the sequence obviously leads to information loss. The author also mentioned the shortcomings of the averaging method, but did not use a better aggregation method
•No method for distinguishing between sensitive and non-sensitive tokens was proposed(The same as the second point)
In many places in the text, the author points out that the traditional sequence-level MIA has shortcomings because it ignores fine-grained member signals, and the signals of non-sensitive tokens may dilute or mask those of sensitive tokens. On page 7, the author points out the deficiency of the AUC metric, that is, AUC may reflect the signal of non-sensitive tokens to a greater extent. However, throughout the entire article, the author did not provide any effective methods for distinguishing sensitive tokens from non-sensitive ones. They merely claimed that users could check by themselves whether a token was sensitive, but this was clearly not from the perspective of an adversary. Therefore, I think the author should provide an automated algorithm to distinguish between sensitive tokens and non-sensitive tokens, making the article more persuasive

Minor issues：
Despite pointing out the limitations and flaws of the AUC metric, the results presented on page 8 show that the attack in this paper achieved a higher AUC value, which sounds somewhat strange

---

> ### Author Response · Authors · 2025-11-24
> **Rebuttal to Reviewer hMRp**
>
> Thank you very much for your review. Here are our responses.
> - > "the method used when migrating from the token-level to the sequence-level in the author's proposed approach is also a common aggregation method. Is this somewhat contradictory"
>
> This is an interesting question, and we believe there may be a misunderstanding. Our work contains two use cases: token-level analysis and sequence-level auditing implemented on top of the token-level framework. While we argue that naive aggregation may dilute sensitive information, the limitation comes from the privacy notion itself rather than from the aggregation operation.
> Sequence-level membership inference is, by definition, an aggregated notion, so any method attempting to audit sequence-level privacy must aggregate token-wise signals and no attack can avoid this. As you note, assuming an adversary knows which tokens are sensitive is unrealistic, which is precisely why sequence-level auditing remains the standard threat model.
>
> Our contribution is that the token-level framework supports both modes:
> - When the adversary has no prior knowledge, our token-level signals can still be aggregated to obtain a strictly stronger sequence-level attack than RMIA.
> - When the adversary has token-specific knowledge, our method enables fine-grained visualization of leakage at each generation step.
>
> Thus, there is no contradiction: the aggregation is required by the definition of the sequence-level notion, and our token-level method simply provides a strictly more informative basis (signal space vs output space) from which aggregation can be performed.
>
> - > "Lack of a mechanism to distinguish tokens from the opponent's perspective"
>
> We understand the concern regarding how an adversary would identify sensitive tokens. However, this ability is part of the threat model, not part of the attack algorithm itself, and we will clarify this explicitly in the paper. As we mentioned above, our token-level framework can be used to audit the sequence-level notion, or conduct more fine-grained analyze on the token-level. If the adversary has no knowledge of what specific tokens to focus on, he can only go for the first option. However, if he does know what tokens to pay attention to, he can go for the token-wise analyze. Hence, the ability to distinguishing private tokens is part of the threat model, not part of the attack. The token-level information leakage is not designed as a global, quantifiable metric that one can compute on a dataset. It is intended to be a inspection tool for auditors to check for precise locations where information leakage happens. An example where the token-level analysis makes sense is that I am checking if the target model memorizes my private information in a given text. I, as the data owner, know what are sensitive in the given text, so I can check the scores on those important tokens. On the other hand, automatically inferring sensitivity of tokens is a separate research problem, orthogonal to membership inference and maybe closer to linguistic or semantic privacy classification, as the sensitivity can be influenced by the context (as another reviewer points out): a name is private in patients' database, but not in the authors section of a paper.
>
> Hence, we are not claiming we have proposed a new metric that quantifies privacy here. We propose a tool that can audit privacy according to existing notions better and provide more fine-grained results.
>
> - > "the results presented on page 8 show that the attack in this paper achieved a higher AUC value, which sounds somewhat strange"
>
> We also presented TPR at small FPR values. These are the only metrics where attack performance can be evaluated on. For a complete evaluation, we included both numbers in the paper. Again, although the sequence-level notion is "lossy", without additional adversary knowledge, it is the only quantifiable notion to use.
>
> I believe your last point is also along the same way as the previous two questions. I hope my answers to your questions have addressed your concerns. In simpler words, there are two complementary use cases of our token-level framework:
> Our token-level framework serves two distinct purposes:
> 1.	A sequence-level privacy auditor when no knowledge about sensitive tokens is available (the standard MIA setting).
> 2.	A diagnostic tool for data owners or auditors who do know which tokens are sensitive and wish to verify whether their private information has been memorized.

---

### Official Review · Reviewer_Kseq · 2025-11-01

**Soundness:** 3
**Presentation:** 4
**Contribution:** 3
**Rating:** 6
**Confidence:** 3

**Summary:**

This work presents InfoRMIA an information theory centric method for membership inference. There are two key contributions: a continuous test static which is superior to a discrete statistic that RMIA uses. This improves AUC and TPR, at low FPR. The second contribution is a token level member inference framework. This is distinct from currently used sequence level methods. The results with a token level framework show that sequence level scores are not accurate due to conflation of private and non-private tokens within a sequence.

The work shows that this method outperforms RMIA and is less sensitive to population size. In addition the token level membership inference results that are stronger than sequence level membership inference. They especially point to sequences which are memorized with no private tokens over estimating the privacy risk, while also pointing to the challenges of using AUC as a privacy metrics

**Strengths:**

1. Better test statistic: Current work reframes RMIA’s setup as a composite hypothesis test, propose a log-likelihood ratio that decomposes into per-sample info gain term and a KL divergence term. This decomposition into memorization and generalization term provides a good interpretability/diagnostic lens,  importantly eliminates gamma, and removes sensitivity of populations set Z.
The derivation is sound and statistically rigorous. Computationally efficient and avoids scaling with large population sets.

The equation turns is smoother, information theory centric relaxation of RMIA equation, there by turns discrete counting into a counting BayesFactor score. This makes sense conceptually and is mathematically sound.

2. Empirically the work shows that this improves AUC and TPR, at low FPR on tabular, image and text datasets. The overall results show that this superior to RMIA.

3. Token level MIA enables a more granular signal and thereby enables more precise decision making related to membership inference. The work also argues and shows sequence level MI conflates private and non private tokens. Thus token level MIA yields a stronger MI.

**Weaknesses:**

1. The KL term assumes normalized distribution over data. However details around estimator or smoothing do not seem fully specified.  This could impact the KL term i.e. how sensitive is KL term to these choices

2. RMIA seems to be the baseline comparison. Are there other MIAs with which the model can be compared to make SOTA claims?

3. Token to sequence aggregation: performance depends on type of aggregation like min-k, mean. While the paper acknowledges this it will be good to show additional aggregators like trimmed mean and also try tuning free calibrations to reduce dependence on min-k

**Questions:**

1. How is p(z/theta) instantiated over the LLM implementations. Is this treated as an empirical distribution or logits over some token set? Please provide details around normalization, smoothing and detail how is KL term impacted by these?

2. What is the runtime overhead when comparing sequence level methods on longer inputs and large vocabularies?

3. For MIMIR experiments how sensitive are results to an early checkpoint, domain mismatch between reference and target?

4. Would it be possible to include other reference based MIAs to support SOTA claims?

---

> ### Author Response · Authors · 2025-11-24
> **Rebuttal to Reviewer Kseq**
>
> Thank you for your review. Here are our responses.
> - > [Q1][W1]"how sensitive is KL term to these choices" "How is p(z/theta) instantiated over the LLM implementations."
>
> The KL divergence form is established as a equivalent form of the test statistic, and the equality only holds when $p(z)$ and $p(z|\theta)$ are probabilistic distributions. Hence, we mentioned that normalization is needed to get to the KL form. However, in implementation, we did not use the KL form to compute the test statistic. We use the second last line of Eqn 3 and Eqn 8, which are more efficient and closer to the original implementation of RMIA. We will clarify this in the implementation details.
>
> To compute $p(z|\theta)$, we use the standard way of treating the softmax scores as the likelihood. In token-level InfoRMIA, all "wrong" tokens in each token generation step in the vocabulary are treated as $z$. For example, if the vocabulary is {0,1,2,3}, and the ground truth $x$ is 2, then $Z=\{0,1,3\}$. Since the softmax scores add up to 1, including $x$, hence we emphasized that in Equation 6-9 that there is an alterative formulation that does not require normalization. We will clarify this in the paper.
>
> - > [Q2] runtime overhead
>
> InfoRMIA has no asymptotic overhead beyond one forward pass per sequence. In a forward pass, at each token generation step, the softmax scores over all possible tokens in the vocabulary will be computed. We take the softmax score of the ground-truth token as $p(x|\theta)$ and take the rest as $p(z|\theta)$. No additional computation is needed, unlike RMIA that requires computing signals on a large population dataset. We will add pseudocode of RMIA, InfoRMIA and token-level InfoRMIA in the paper to better explain the attack algorithm and demonstrate the improved computational efficiency.
>
> - > [Q3] "how sensitive are results to an early checkpoint, domain mismatch between reference and target?"
>
> We will add 9 additional tables to the paper in the appendix to show the performance when we use later checkpoints as the reference model. The results agree with our argument for choosing step 1 checkpoint in Line 452: the later the checkpoint, the less OUT the reference model is, the worse the performance.
>
> MIMIR and Eleuther (creators of Pythia models) do not provide out-of-domain reference models, but we believe the step-1 checkpoint effectively behaves like an out-of-domain snapshot because it has seen only a small number of documents. Based on the RMIA paper, we expect that genuinely out-of-domain reference models would reduce attack performance somewhat, but not drastically.
>
> - > [Q4][W2] "include other reference based MIAs to support SOTA claims"
>
> Yes—we will update Tables 1 and 2 to include LiRA. Below is a preview of the updated Table 2. Initially, we omitted LiRA because RMIA already demonstrated its dominance over all prior methods, but we agree it is better to include LiRA for completeness and clarity.
>
> | Dataset        | FT Epochs | RMIA AUC  | RMIA TRP@FPR | InfoRMIA AUC | InfoRMIA TRP@FPR | InfoRMIA (token) AUC | InfoRMIA (token) TRP@FPR | LiRA AUC | LiRA TRP@FPR |
> | -------------- | --------- | --------- | ------------ | ------------ | ---------------- | -------------------- | ------------------------ | -------- | ------------ |
> | **AG News**    | 1         | 0.839     | 0.00%        | **0.843**    | **23.0%**        | 0.836                | 20.2%                    | 0.795    | 3.6%         |
> |                | 4         | **0.945** | 0.00%        | **0.945**    | 16.2%            | 0.942                | **20.6%**                | 0.882    | 9.00%        |
> | **ai4privacy** | 1         | 0.643     | 6.6%         | **0.644**    | **10.6%**        | 0.620                | 9.0%                     | 0.630    | 3.8%        |
> |                | 4         | 0.821     | 26.0%        | **0.822**    | **27.2%**        | 0.804                | 23.2%                    | 0.782    | 10.4%         |
>
> - > [W3] "it will be good to show additional aggregators like trimmed mean and also try tuning free calibrations to reduce dependence on min-k"
>
> We tried multiple aggregators: mean, median, trimmed mean, one-sided trimmed mean, min-k, max-k, and some black box aggregators. However, no one can consistently beats averaging, including the black box ones. We agree that tuning-free aggregators should be favored, so we mostly used averaging in the paper. We can include additional aggregation results in the appendix for interested readers. Thank you for the suggestion.

---

### Official Review · Reviewer_Vt48 · 2025-11-01

**Soundness:** 2
**Presentation:** 2
**Contribution:** 2
**Rating:** 4
**Confidence:** 4

**Summary:**

This paper presents InfoRMIA, a membership inference attack (MIA) designed specifically for large language models (LLMs), specifically based on the existing RMIA framework. Then the authors argue that sequence-level MIAs are fundamentally flawed for LLMs, as the privacy risk is localized to individual tokens rather than entire sequences. To address this, they propose a token-level privacy assessment framework that can more accurately identify privacy leakage at the token level.

**Strengths:**

1. Membership inference attack on large language models is a critical area of research.
2. The proposed InfoRMIA method is effective.

**Weaknesses:**

1. The organization of the paper could be improved, particularly in providing sufficient context for RMIA and clearly defining the threat model.
2. The threat model needs further discussion, especially regarding the assumption of having access to a dataset drawn from the same distribution.
3. The idea of token-level membership inference needs further discussion, particularly regarding the definition of privacy leakage.
4. Experimental results in Table 1 are hard to interpret.

**Questions:**

1. The organization of the paper could be improved. Specifically, Section 3.1 briefly introduces RMIA, without much detail, and even the problem statement and threat model are not clearly defined. The authors directly jump into the technical details of InfoRMIA without providing sufficient context. RMIA is a relatively new method, and readers may not be familiar with it, even those who are well-versed in membership inference attacks.

2. Another issue linked to the unclear threat model is that it is not clear what the attacker's capabilities are. For example, I checked Appendix A, and saw RMIA requires a dataset that is drawn from the same distribution. Does InfoRMIA have the same requirement? And although this assumption is common in traditional MIAs, it needs further discussion in the context of LLMs. Specifically, for traditional MIAs, the models are usually trained on relatively small and homogeneous datasets, like training images on CIFAR-10. So it's clear what "the same distribution" means. However, LLMs are typically trained on massive and diverse datasets, making it less clear what "the same distribution" means in this context. For example, if we want to infer whether a medical text is part of the training data, are we supposed to sample other medical texts from the same distribution as the target text? And many sensitive data are quite rare, making it difficult to find similar data points from the same distribution. Therefore, I would like to see a more detailed discussion on this assumption and its implications for the practicality of InfoRMIA.

3. The idea of token-level membership inference needs further discussion. I agree that certain tokens in a sequence may pose greater privacy risks than others. However, the definition of privacy leakage for individual tokens requires further discussion. The privacy risk of a token are also dependent on its context within the sequence. For example, the name of the patient in a medical report may be sensitive, but if it is appears in an author list of a research paper, it may not be sensitive at all. Similarly, a password is highly sensitive if it's surrounded by the corresponding login information, but if it appears in a random sentence, it may not be sensitive. Therefore, claiming sentence-level MIAs are fundamentally flawed may be too strong without further discussion.

4. Experimental results in Table 1 are hard to interpret. I would expect that sampling more data would lead to better attack performance, and it's true for RMIA. However, for InfoRMIA, the attack performance remains identical even to four decimal places. It would be helpful to provide some explanation for this phenomenon.

---

> ### Author Response · Authors · 2025-11-24
> **Rebuttal to Vt48**
>
> Thank you for your review. Here are our responses.
> - > [Q1][W1] "The organization of the paper could be improved"
>
> Thank you for your suggestion. We will move the description of RMIA from the appendix into the main text and expand it for a smoother reading experience, especially now that we have an additional page available at this stage of the submission.
>
> - > [Q2][W2] "not clear what the attacker's capabilities are."
>
> Thank you for pointing it out. We will add a problem statement section that describes the threat model, and provide a pseudocode. Hope these additions can address your concern.
>
> You are right that RMIA needs an additional population dataset that is of the same data distribution. This assumption is relatively valid under traditional ML or finetuning LLM settings where we have validation/test datasets where we can split a population set. One big contribution of InfoRMIA is the **reduction the dependence of the population set**, which we have proven theoretically and empirically. With pretrained LLMs, we recognize it is almost impossible to get additional population set, so we introduce **token-level InfoRMIA that requires no population data**. This makes the attack very practical, hence a huge contribution to the field. Note that the MIMIR results in the paper are obtained with this token-level InfoRMIA that requires no population data.
>
> ### Algorithm 1: MIA Score Computation (RMIA / InfoRMIA / Token-Level InfoRMIA)
>
> Input:
>     - Target model θ
>     - Reference models Θ (size k)
>     - Query x
>     - Hyperparameters γ, a
>     - Population dataset Z (for RMIA and InfoRMIA)
>
> Output:
>     - MIA score Score_MIA(x; θ)
>
> ````python
> 1:  Compute p(x|θ) and p(x|θ_r) for all θ_r in Θ
>
> 2:  p_out(x) ← (1/k) * sum_{θ_r in Θ} p(x|θ_r)
>
> 3:  p(x) ← 0.5 * ((1 + a) * p_out(x) + (1 - a))
>
> 4:  Ratio_x ← p(x|θ) / p(x)
>
> 5:  if Token-Level InfoRMIA then
> 6:       Take all non-ground-truth tokens as z        (See Sec. 4.2)
> 7:  else
> 8:       Take population samples z from dataset Z
> 9:  end if
>
> 10: for each z do
> 11:      Compute p(z|θ) and p(z|θ_r) for all θ_r in Θ
> 12:      p(z) ← (1/k) * sum_{θ_r in Θ} Pr(z|θ_r)
> 13:      Ratio_z ← p(z|θ) / p(z)
> 14: end for
>
> 15: if RMIA then
> 16:      Score_MIA(x; θ) ← (1/|Z|) * sum_z I( Ratio_x / Ratio_z ≥ γ )
> 17: else
> 18:      Score_MIA(x; θ) ← log(Ratio_x)
>                           − sum_z p(z) * log(Ratio_z)
> 19: end if
> ````
>
> - > [Q3][W3] "The privacy risk of a token are also dependent on its context within the sequence"
>
> You are absolutely right that the sensitivity of a token depends on its context. This is exactly what our method captures. The term “token-level analysis” may sound like it refers to individual token identities, but in fact it refers to each token generation step, which inherently incorporates context.
>
> As we explain in Section 4.1, each token generation step takes a prefixal subsequence as input and outputs a token. The membership score of a token is therefore the likelihood of producing that token given its prefixal context. For example, for the sequence “My name is Bob,” token-level membership scores come from: p(name|My), p(is|My name), p(Bob|My name is). Thus, token-level InfoRMIA naturally incorporates contextual information and is not a context-free notion of privacy.
>
> - > [Q4][W4] "Experimental results in Table 1 are hard to interpret"
>
> Very sharp observation. We will definitely add the explanations to the paper.
>
> The same numbers from InfoRMIA are not errors. Table 1 is done under the setting of auditing the privacy risk of **a fixed target model**. If you look at Eqn 3, which is the test statistic of InfoRMIA, the second term is independent of $x$. Thus, in this case, the ranking of the test statistics of all test samples $x$ will not be affected by $z$, because the second term is the same for all $x$. Therefore, increasing $|Z|$ will have no effect on the performance of InfoRMIA. However, for RMIA, the test statistic is a percentile computed on $z$, thus it gets more precise with a larger $Z$. An extreme case is that when $|Z|=1$, the RMIA's test statistics will always be 0 or 1, leading to bad performance. With a larger $Z$, the percentiles can take on more diverse values, and thus more precise. This is why RMIA benefits from a larger $Z$ while InfoRMIA does not in this setting.
>
> If you are wondering why we keep the second term in Eqn 3, this is due to cases where we want to audit privacy according to a different privacy game. Ye et al. (2022) formulate and categorize multiple privacy games. Particularly, when we want to audit privacy of a training algorithms, we need to train multiple models and aggregate their outputs. In this case, the second term is no longer the same for all $x$ as there are multiple $\theta$.
>
> In summary, your observation is very accurate, and we will explain the reason in more depth in the paper. Hope we have address all your concerns.

---

### Author Response · Authors · 2025-11-25
**Paper updated**

Dear reviewers

We have updated the paper with all of your constructive feedback. Here, we outline the major changes to the paper.

 With the extra page, we now include a problem statement of the membership inference and a more detailed explanation of RMIA in the main text. We also clarified the confusing parts raised by some of you, and expanded some relevant sections based on your reviews. In the experiment section, we added LiRA as an extra baseline to substantiate our SOTA claim. In the appendix section, we included a consolidated pseudocode for easier comparison between RMIA, InfoRMIA and token-level InfoRMIA. We also explained more experiment setup and included additional experiment results on MIMIR when using later checkpoints as reference models.Furthermore, we added a GPT-Neo experiment on MIMIR for interested readers, whose reference models are completely IN. This supports our claim that earlier checkpoints are a better choice of reference models when attacking pretrained LLMs. We also made various minor improvements to the writing and overall organization of the paper.

We hope this revision has addressed all of your concerns. We will continue to improve our paper if you have further suggestions.

---

### Meta-Review · Area_Chair_Hcgw · 2025-12-30

**Summary:**

The paper proposes InfoRMIA, an extension of RMIA which leverages token-level signals for the attack. This is then used to identify tokens with highest privacy risk, but also to get a strong sequence-level MIA. The reviewers' concerns were properly addressed during the rebuttal, see the following field.

**Reviewer Concerns:**

The first reviewer had some concerns regarding the assumptions and threat model, which have been sufficiently addressed in the rebuttal. The same holds for their concern on the context of private information (e.g., a name on a paper is non-private, while it is private on a patient record), where the authors clarified that the autoregressive setup can capture exactly this.

-> would probably have increased their score

The second reviewer was concerned about the KL-divergence and assumption on normality, which has been clarified by the authors (by stating it was not used for the computation). Also the concern about runtime overhead was addressed: overhead seems to be low (one autoregressive forward pass). Additional experiments provided by the authors also assessed the choice of checkpoint and the integration of LiRA.

-> would probably have increased their score

The third reviewer expressed concerns about the identification of private tokens. I will come back to this point in my own review, using reference [R1]. The authors explain that this is part of the threat model and not a limitation of the method itself, and this seems fair. They also ask for more effective aggregation functions. Given that the paper shows already strong competitive results with the „naive“ aggregation, this seems like something to optionally explore in future work on further improving the application of the method on sequence tasks. In a similar vein, the described „concealment“ does actually not seem to be concealment but rather the loss of the method on sequences when the adversary has no further knowledge.

-> might not have increased their score due to conceptual disagreement with the setup of the paper, but I do think that the concerns, while good to be addressed in writing, are not diminishing the contributions of the paper

The fourth reviewer’s concern about computation overhead is equally addressed. Their concern that actually private tokens have low privacy score. The authors argue that this is due to the nature of the private tokens, which are „outliers“ and not sufficiently learned my the model. This seems to be a true limitation of the method which should be more prominently discussed. Based on the provided updated paper, I was not able to identify where the discussion that the authors promised in the rebuttal was included. A concrete pointer here would have helped me, so I am putting it on the Todo-list for the authors to make that more prominent. Moreover, the authors’ argument that DP would be a good protection is valid, and it does not seem absolutely necessary to require additional experiments here. We would expect drops in attack success. The concern about too little experiments has been addressed by adding also experiments on GPT-Neo, and going through the appendix, I actually found experiments to be abundant.

-> should have probably increased their score

The final reviewer expresses concern regarding the clarity and presentation of the theoretical derivations and the experimental setup. While the theoretical concerns are properly addressed, the experimental setup is still unclear partially, see AC review [R2]. The concern regarding the unstability of the method was addressed by pointing to the difficulty of MIA on those datasets which seems valid. The addition of Lira addressed another concern. Finally, also the concern about performance between InfoRMIA and RMIA were explained by the fact that it was shown that InfoRMIA cannot underperform RMIA.

-> expressed their desire to increase the score to 8.

**Reviewer Scores:**

As indicated above, it is likely that at least four out of the five reviewers would have increased their scores, given that their concerns were fully addressed.

**The AC’s review and comments**

In addition to the reviewers input, the AC also carefully reviewed paper and appendix for a more reliable assessment. The points below summarize the AC’s take.

- [R1] the AC’s main concern regarding the private tokens is not about the choice of private tokens in the threat-model, but the fact that it is very much unclear what values and results we are looking at in the first half of the paper regarding the private tokens:
**1.** The paper shows that the correlation between MIA on the whole sequence and MIA and the private tokens is weak (Section 5.1), and reference to Figure 1a. But what was used to select the private tokens? What dataset was used in the first place? What model was used? Please clarify this setup both in Section 5.1 and in the respective captions to make the paper self-contained.
**2.** In a similar vein, the paper also  discovers that many of the most “memorized” sequences either contain disproportionately little private information (Figure 7) or no private information at all (Figure 2). Also for figure 2 and 7 and this part, it is completely unclear what results are depicted and this needs to be fixed. And
**3.**, ([R2]) the experimental description could be still further extended. For example: How is AGNews used for „generation“ rather than classification? Appendix F1 only „In our experiment, we ignore the labels column and train autoregressive models on it.“  This is too little. Is the output truncated? How long are the generated sequences?
- One other thing that remains unclear to me why, and more importantly *how* does the paper evaluate CIFAR10? Where are the „tokens“ here? This is currently not explained. Could the authors please add another explanation and maybe pseudo-code to clarify how such a token-based approach can be applied to data that clearly does not have tokens.
-  Finally, looking at Table 1, I found that to be very little expressive in terms of the choice of size  Z. F None of the methods has hardly any  difference between 1000 and 10000. Maybe it would make sense to just try and add more extreme values for that table to see actual differences in the method? I suppose the message is supposed to be that InfoRMIA is more robust to the choice of (smaller) Z?

**Minor revision list for the authors:**

Based on the successful rebuttal in addressing all reviewers’ concerns and the contributions by the paper, I recommend an accept. However, there is additional work that needs to be done for the paper to serve the community properly. It seems uncommon for an AC to give authors a revision list, but under the given circumstances where we could not interact during the rebuttal phase, this seems like a viable option to recognize the paper’s value but still straighten last questions. The following represents the AC’s minor revision list for the authors to perform upon acceptance.

- Add prominently to the paper the discussion that certain types of private tokens (like passwords or other more outlier strings) might actually get a too low privacy
- Clarify all 3 points regarding [R1]
- Clarify the setup for running InfoRMIA a non-token data like CIFAR10 or tabular data and explain where the gains there stem from
- Extend Table 1 to values where we would actually see a difference between the different methods

---

### Decision · Program_Chairs · 2026-01-26

Accept (Poster)